# Selection and horizontal gene transfer underlie microdiversity-level heterogeneity in resistance gene fate during wastewater treatment

Connor L. Brown[1], Ayella Maile-Moskowitz[1], Allison J. Lopatkin[2], Kang Xia[3], Latania K. Logan[4], Benjamin C. Davis[5], Liqing Zhang[6], Peter J. Vikesland [1,7] ✉ & Amy Pruden [1,7] ✉

Activated sludge is the centerpiece of biological wastewater treatment, as it facilitates removal of sewage-associated pollutants, fecal bacteria, and pathogens from wastewater through semi-controlled microbial ecology. It has been hypothesized that horizontal gene transfer facilitates the spread of antibiotic resistance genes within the wastewater treatment plant, in part because of the presence of residual antibiotics in sewage. However, there has been surprisingly little evidence to suggest that sewage-associated antibiotics select for resistance at wastewater treatment plants via horizontal gene transfer or otherwise. We addressed the role of sewage-associated antibiotics in promoting antibiotic resistance using lab-scale sequencing batch reactors fed field-collected wastewater, metagenomic sequencing, and our recently developed bioinformatic tool Kairos. Here, we found confirmatory evidence that fluctuating levels of antibiotics in sewage are associated with horizontal gene transfer of antibiotic resistance genes, microbial ecology, and microdiversity-level differences in resistance gene fate in activated sludge.

Wastewater treatment plants (WWTPs) have been referred to as "hot-spots" for the proliferation and dissemination of antibiotic resistance[1]. However, this is a broad generalization, and there is a need to identify more precise circumstances and boundary conditions under which such proliferation occurs. In fact, the activated sludge (AS) process that serves as the core of conventional wastewater treatment can be quite effective at reducing antibiotic resistance gene (ARG) numbers at-large[2] and has been noted to be generally effective at attenuating certain mobile ARGs, especially those carried by fecal pathogens in the influent sewage[3,4]. Still, there are a number of ARGs that have been observed to sometimes increase during wastewater treatment, thus

calling for a closer examination of specific vulnerabilities of the AS process to ARG proliferation[5]. Horizontal gene transfer (HGT) is of particular concern because it can result in the acquisition of ARGs across taxonomic groups, resulting in the emergence of new resistant bacterial strains. WWTPs have been found to be susceptible to invasion by exogenous mobile genetic elements (MGEs)[6] and also to harbor diverse mobile resistance genes[3,7,8], including both putative novel ARGs as well as MGEs and bacterial hosts associated with ARG emergence. Despite this, concrete examples of selection for resistance via HGT in situ, i.e., within a fixed- and measured period of time in AS, are surprisingly sparse[9,10].

[1]Dept. of Civil and Environmental Engineering, Virginia Tech, Blacksburg, USA. [2]Dept. of Chemical Engineering, University of Rochester, Rochester, USA. [3]School of Plant and Environmental Sciences, Virginia Tech, Blacksburg, USA. [4]Dept. of Pediatrics, Emory University, Atlanta, USA. [5]Office of Research and Development, U.S. Environmental Protection Agency, Cincinnati, USA. [6]Dept. of Computer Science, Virginia Tech, Blacksburg, USA. [7]These authors jointly supervised this work: Peter J. Vikesland, Amy Pruden. ✉e-mail: pvikes@vt.edu; apruden@vt.edu

HGT is a stochastic process and can co-occur via complex ecological interactions in microbiomes, making it difficult to study under controlled conditions. In vitro, various pharmaceuticals[11] and sterile-filtered hospital effluent[12], have been shown to elevate conjugation rates, a key mechanism by which ARGs are hypothesized to proliferate during wastewater treatment. Similarly, whole genome sequencing of WWTP isolates has served to demonstrate linkages between WWTP microbes and human/animal bacteria[13] despite no selection for multi-drug resistance overall due to biological wastewater treatment[14]. Other efforts using shotgun metagenomics have provided high-level surveys of putative gene sharing and the potential for ARG mobility as a function of co-occurrence with MGEs[15] or correlation between ARG and MGE abundances[16,17]. The correlation of ARG and MGE abundances through short-read mapping is particularly problematic, as changes to gene abundance are driven largely by changes in the abundance of the host bacteria. By contrast, HGT occurs between individual cells, suggesting that a much greater degree of biological granularity is required to reveal HGT and corresponding drivers in WWTPs. Likewise, the degree to which in vitro studies are representative of the complex and dynamic microbiomes that typify WWTPs, particularly AS, is questionable. Studies leveraging model organisms or single modes of HGT (e.g., conjugation or transformation) under controlled conditions are also unlikely to capture dynamics among environmental and mostly unculturable[18] AS taxa. We recently proposed a framework of "in situ" HGT that assesses whether the chronological occurrence of potential donors, recipients, and putatively transferred regions could plausibly have arisen due to HGT in the sampled period. To develop this framework as a publicly accessible tool, we recently developed Kairos[19] as a next-flow software package that enables analysis of metagenomes for evidence of microbiome-level HGT.

Studies of controlled situations where resistant bacteria, ARG-bearing MGEs, and selective agents are elevated in the influent to the WWTP could help to shed light on the microbial ecology surrounding ARG proliferation in AS[20]. In particular, relative to municipal wastewater, hospital sewage tends to be enriched with microbial pathogens, ARGs, and selective agents, such as antibiotics and other pharmaceuticals, which could enhance both HGT and selective pressure on resistant strains[21–23]. Such studies could help to assess the utility of mitigation measures, such as segregating hospital sewage or subjecting it to special treatment prior to discharge. Typically, hospital sewage constitutes only a small proportion (0.01%–15%[24–28]) of the total influent reaching municipal WWTPs, but the potency of this "small" proportion in terms of antibiotic resistance propagation remains a concern.

Here, we employed sequencing batch reactors (SBRs) for a semi-controlled simulation of AS wastewater treatment and allowed a comparison of the effects of varying influent conditions on the HGT of ARGs occurring during AS treatment. The SBR feeds varied as a function of contrasting levels of influent hospital sewage composition (0% and 10%) and also natural variation in the selective agents present in the municipal sewage source applied as influent with time. SBRs simulating AS treatment were ideally suited for the in situ study of HGT of ARGs because they are representative of the complexity of microbiomes encountered in full-scale WWTPs but can be operated in parallel and in triplicate to account for the influence of biological variability. To profile key networks between/within phylum HGT, we applied Kairos as a means to leverage microdiversity-aware sequence analysis for sensitive detection of microdiversity in gene contexts associated with HGT[19]. This approach served to identify evidence of HGT that was linked to shifting antibiotic levels. The framework used herein can serve as a template for profiling HGT in longitudinal metagenomic datasets and highlights that the AS microbiome remains an important focal point for efforts to monitor and mitigate the spread of antibiotic resistance.

## Results

### Overview of experimental design

Six replicate SBRs were operated with local municipal sewage as feed until they reached a steady state (defined in this study as stable removal of organic carbon) (~3 months) (Fig. 1A). Subsequently, hospital sewage was blended into the influent to one set of biological triplicate SBRs at a proportion of 10% hospital effluent to municipal sewage. Given that concentrations as high as 15% of hospital sewage have been reported[28], 10% was selected as the test condition to maximize the chance of observing the impact of hospital sewage on AS. Sampling was carried out for short- and long-read metagenomics and suspect screening of pharmaceuticals and personal care products (PPCPs) over a period of about three weeks. AS and influent samples were sequenced to an average depth of 5 Gbp/sample (nonpareil[29] coverage $0.5 \pm 0.1$) and effluent was sequenced to an average depth of 3 Gbp/sample (nonpareil coverage $0.5 \pm 0.1$). A subset ($n = 6$) of samples were sequenced deeply (average of 36 Gbp, nonpareil coverage $0.8 \pm 0.10$). A subset of DNA extracts from biological replicate reactors was also pooled by sampling date and sequenced across three nanopore minION flowcells to a target depth of 1.2 Gbp/sample and 9.4 Gbp total after basecalling (sequencing read $N_{50} = 1.3$ kbp) (Supplementary Data 1). The low $N_{50}$ values of the nanopore data reflect the use of a bead-beating lysis DNA extraction kit.

### Limited impact of hospital sewage on activated sludge reactors

Consistent with the results of a companion study focused on relating SBR operational conditions to the higher-level annotation of ARGs and taxonomy[30], hospital sewage was found to have only a minor impact on the organic carbon and nitrogen removal and the composition of the corresponding microbiomes and resistomes (Fig. 1A–D). Short read-derived taxonomic profiles were analyzed at the genus level because pairwise Bray-Curtis distances suggested that conditions (10%/0%) and fractions (i.e., influent, AS, and final effluent (FE)) differed most substantially at this level of taxonomic resolution, even when removing sporadic or low abundance taxa that were potential false positives (< 0.1% abundance in more than half of samples; Supplementary Figs. 1 and 2). However, genus-level taxonomic profiles differed only slightly among treatments when controlling for sampling day, reactor, and fraction (PERMANOVA, $R^2 = 0.015$, $p = 0.001$), influent samples grouped by 0% vs. 10% when controlling for experimental stage (before, initial, and after introduction of hospital sewage) and with no other grouping variables ($R^2 = 0.17$, $p = 0.01$).

*Post hoc* Wilcox Rank-Sum tests were performed to identify taxa with statistically significant differences between the 0% or 10% conditions (Fig. 1D, Supplementary Fig. 3). Despite a difference in taxonomic profiles, no specific genera with statistically significant differences between the influent sewage with or without hospital effluent could be detected (controlling for sampling period, p-value adjusted for multiple comparisons using the Holm method)[31]. The resistome was strongly linked to genus-level taxonomic profiles (Procrustes: $m^2 = 0.90$, $p = 0.001$; excluding undiluted hospital effluent, 0.80, $p = 0.001$), (Fig. 1D, E), suggesting a strong partitioning of ARGs into separate genera.

### Comparing against a custom AS reference genome, ARG, and MGE catalog

We hypothesized that hospital sewage would introduce substantial ARG and MGE microdiversity into the SBRs. To assess this, we developed two genomic catalogs comprising (1) metagenome-assembled genomes (MAGs)/whole genome sequences of isolates recovered from the SBRs or influent and (2) metagenomic scaffolds bearing ARGs or mobileOGs (i.e., MGE hallmark genes) recovered using multiple hybrid assembly, coassembly, and binning strategies. The combined metagenomic assembly yielded over 300 Gbp of hybrid assembly data and

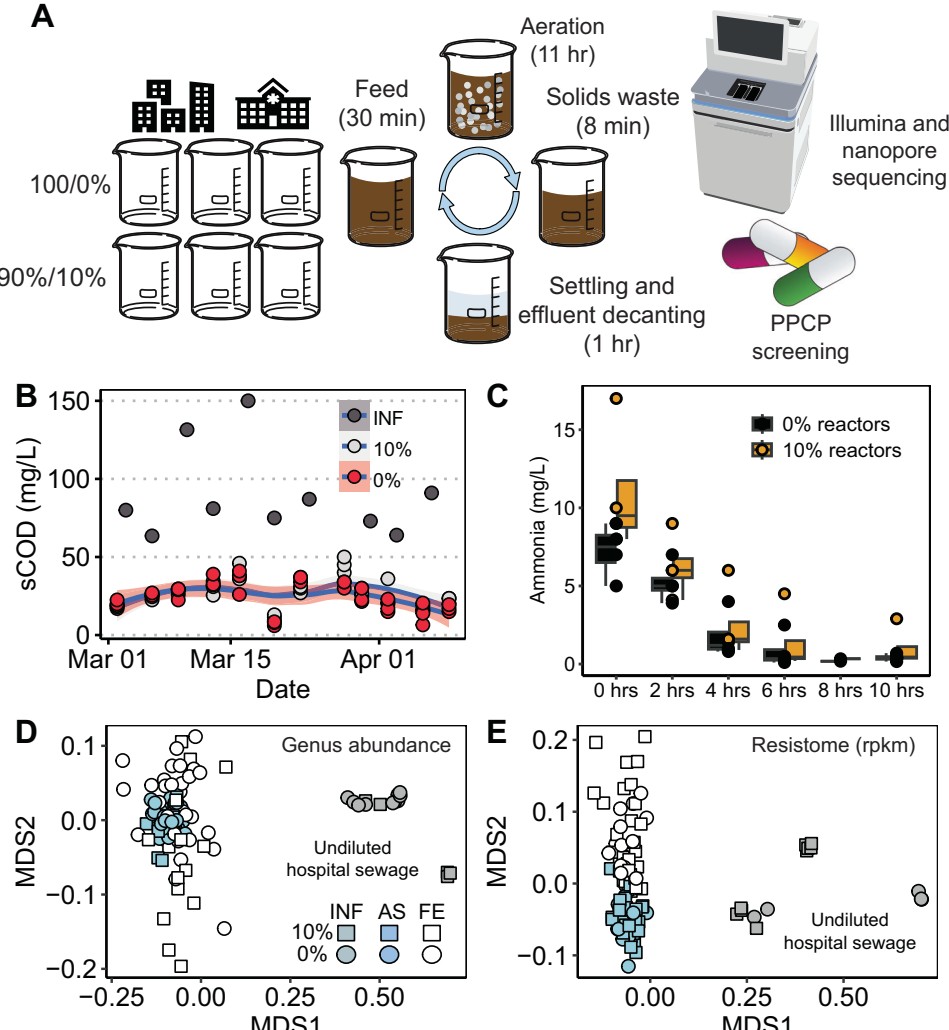

**Fig. 1 | Hospital effluent had modest impacts on the performance, taxonomic, and resistome composition of the sequencing batch reactors (SBRs).**
**A** Experimental design and reactor schematics. A total of six biological replicate SBRs were seeded with a local AS inoculum and upon reaching steady-state were fed three different concentrations of hospital sewage in triplicate. Over three weeks, samples were collected for pharmaceutical and personal care product (PPCP) screening and metagenomics (short and long read). **B** Influent and reactor effluent soluble chemical oxygen demand (sCOD) was not substantially different between 0% and 10% reactors. Lines are a loess curve with standard error bands (**C**) SBRs demonstrated robust nitrification. Data are from three different sampling

days. The time displayed is relative to the beginning of the 11 hr aeration period. Boxplot summary statistics are center line: median; upper/lower hinges: 75th and 25th percentiles, respectively; upper and lower whiskers represent the data points extending from the hinge to at most 1.5 times the interquartile range. **D** NMDS of genus-level taxonomy Bray-Curtis distances between samples of different fractions and treatment conditions (stress = 0.058). **E** NMDS of resistome Bray-Curtis distances between samples of different fractions and treatment conditions (stress = 0.073). Abbreviations: INF: influent; ML: activated sludge; FE: final effluent. Source data are provided as a Source Data file.

after binning produced 876 species-level dereplicated medium- or high-quality MAGs (Supplementary Data 2, Supplementary Fig. 4).

For subsequent analysis of microdiversity and HGT, we relied on the ARG and MGE context catalog derived from the 300 Gbp of hybrid assembly data and subsequently related those findings back to potential hosts via analysis of the MAGs. The final resistance gene and mobileOG catalog consisted of 1,354,363 contigs (total assembly size 1,124,994,800 bp; $N_{50} = 1,587$ bp). From these assemblies, a total of 910 unique reference resistance genes were detected (535 ARGs and 375 metal resistance/biocide resistance genes), which is comparable to previous studies[3].

A diverse array of mobile resistance genes was detected in the MGE and ARG catalogs. Mobile resistance genes were classified according to their co-occurring mobileOGs. The classification scheme was such that individual resistance genes could belong to one or multiple categories of integrative element (IGE), transposable element

(TE), plasmid, phage, or conjugative element (CE). A total of 408 unique mobile resistance genes were detected across 1,544 contigs (Supplementary Fig. 5). Of these, 9 (2%) were co-localized with phage hallmarks, 54 (13%) with TEs; 97 (24%) with IGEs; 108 (26%) with CEs, and 175 (42%) with plasmids. In addition, 79 (19%) were co-localized on contigs with markers for TEs, IGEs, CEs, and plasmids. This latter category of highly mobile resistance genes included OXA-205, *mer* and *qac* genes, *mphE, mphA*, and *aadA*, among others (Supplementary Fig. 5).

Whereas short read profiles suggested a modest difference between hospital- and municipal-sewage, analysis of metagenomic microdiversity via the Kairos assessment workflow revealed 12,675 contigs unique to the hospital sewage-blended feed (Figs. 2, 3). The Kairos assess branch of the next flow pipeline leverages a metagenomic assembly catalog such as the one created here for resistance genes and mobileOGs that spans multiple gene contexts. This

workflow was applied to identify distinguishing regions between gene contexts, extract these regions, and query metagenomic reads against them to determine whether a given contig can be found in a sample. Only 260 contigs were exclusively detected in the background

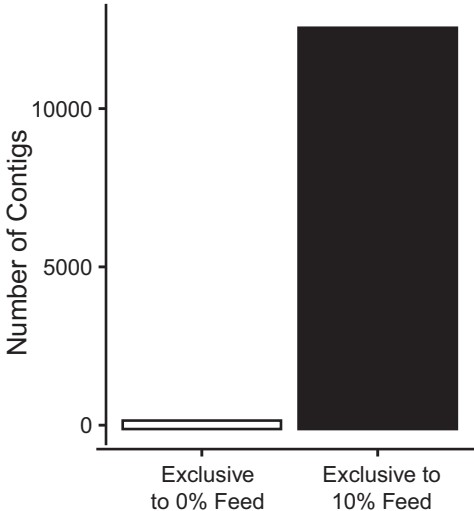

**Fig. 2 | Use of multiple hybrid assembly strategies and the bioinformatics software Kairos predicts a unique contribution to the resistome and mobilome either the 0% or 10% influent.** Contigs are derived from the resistance gene-MGE gene catalog and processed using the Kairos assess workflow. Source data are provided as a Source Data file.

municipal sewage, a strong contrast to the hospital effluent (Supplementary Fig. 6). Of the 12,675 contigs unique to the hospital sewage-blended feed, 232 encoded resistance genes (183 unique), including 31 ARGs and 52 biocide or metal resistance genes that were not detected in 0% hospital sewage influent. For example, the unique contribution of the hospital sewage included 14 contigs encoding macrolide resistance genes *msrE, mphE,* and *tet(39)* in a transposon-like setting (Fig. 3C, D); 39 *sul1*-bearing contigs (Supplementary Fig. 7); 38 *mphA*-bearing contigs among others (Supplementary Data 3). The patterns of abundances predicted by short read to contig alignment concurred with the trends observed using Kairos assess (Supplementary Fig. 8). The overall structure of the mobile resistance gene graphs (Fig. 3A, B) for hospital and native sewage were similar (Supplementary Fig. 9A and B), except that the hospital-associated graphs spanned fewer distinct taxa relative to the background municipal sewage (Supplementary Fig. 9C).

## Microdiversity-level differences in resistance gene fate
The use of the Kairos assess workflow further enabled the partitioning of ARG-bearing contigs into hospital sewage-associated and background municipal sewage-associated fractions (Fig. 4, Supplementary Fig. 11). This partitioning allowed us to trace the fate of specific resistance gene contexts by assessing which hospital or native sewage-associated contigs remained detectable in AS after several days of operation. Both the municipal and hospital sewage-associated resistomes were largely attenuated (Fig. 4) when evaluated using either Kairos (Fig. 4A, B) or reads mapping to the contigs directly (Fig. 4C–E). However, different drug classes, as well as individual ARGs within the

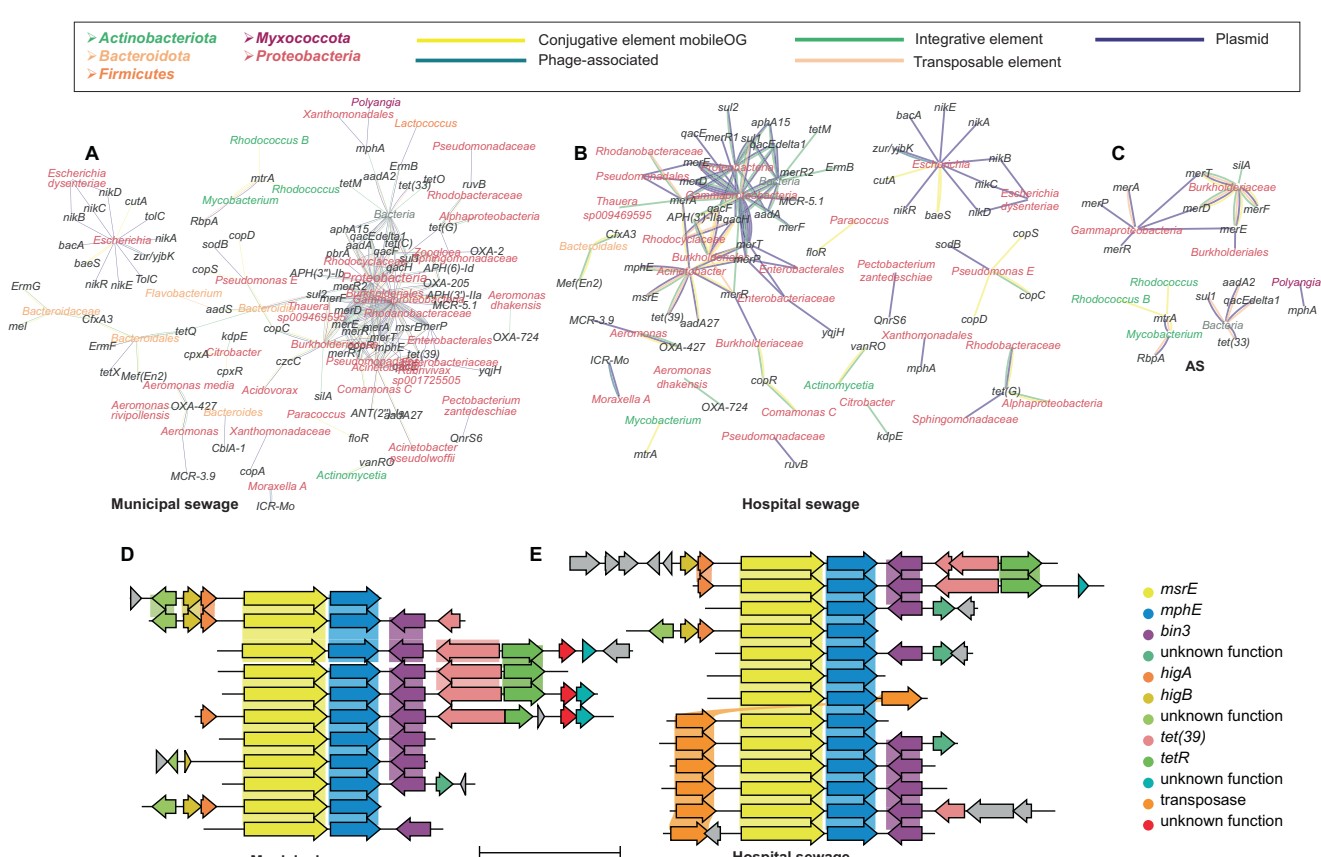

**Fig. 3 | Hospital effluent contributes distinct resistance genes, mobile genetic elements, and bacterial hosts. A** Gene-sharing graph of resistance genes derived from municipal (i.e., native) sewage determined via the Kairos assess workflow. **B** Gene-sharing graph of resistance genes derived from hospital sewage. **C** The innate mobile resistome of AS. **D** Municipal sewage-associated contexts of *msrE/mphE* and *tet(39)* in a transposon-like setting. **E** Same as (**D**), but for contexts present in hospital sewage. (**A**–**C**): edges are mobileOG element classifications, while colored nodes are host taxa (colored by phyla).

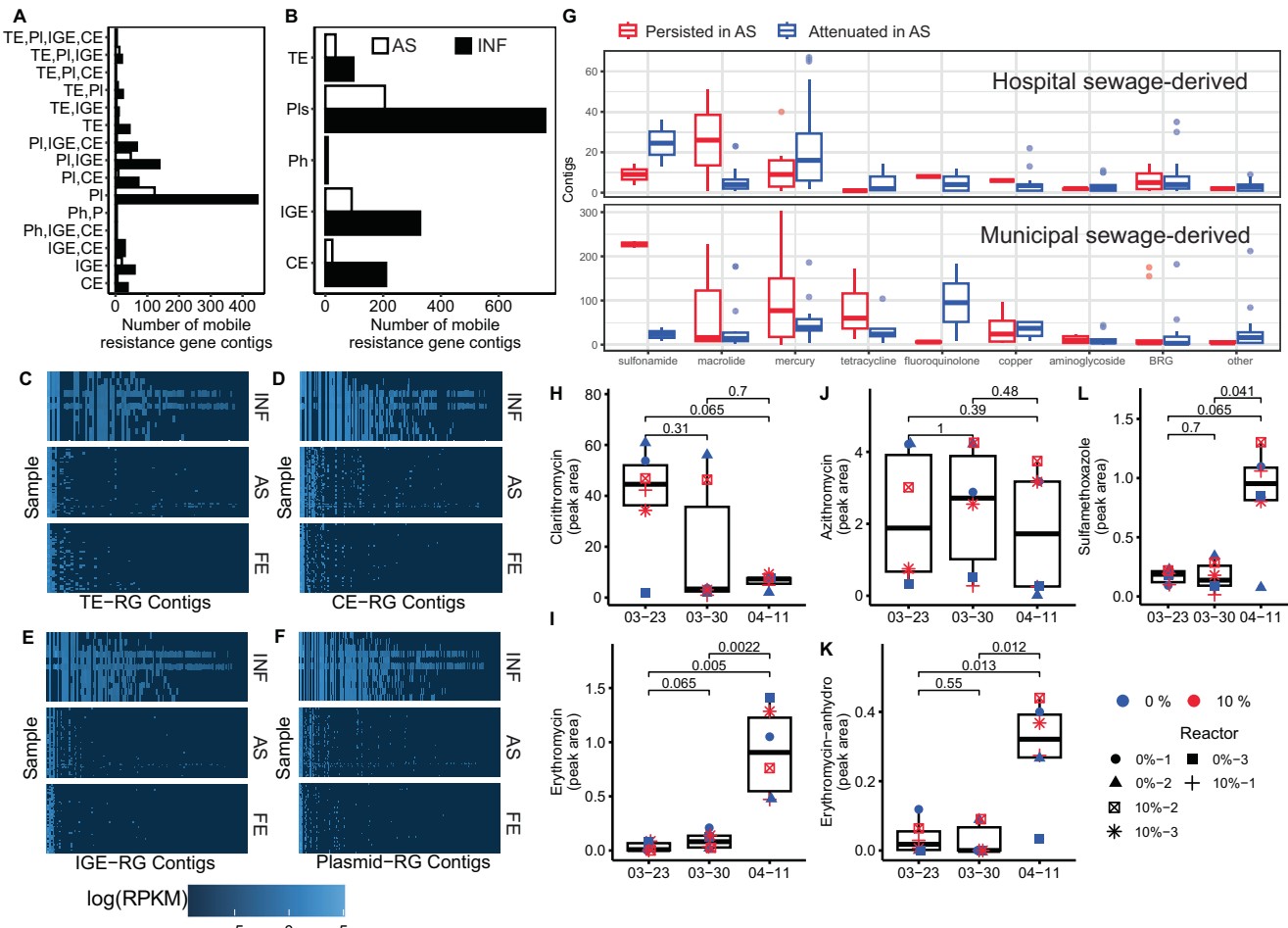

**Fig. 4 | Activated sludge largely attenuated the influent resistome, with microdiversity level differences in resistance gene fate. A**, **B** Mobile resistance genes were attenuated by activated sludge as evidenced by the decreased number of ARG-bearing contigs identified using Kairos assessment. (TE: Transposable element; IGE: integrative genomic element; Pls: plasmids; Ph: Phages). **C**–**F** Heatmap of mobile ARG-bearing contig abundances across influent (INF), activated sludge (AS), and final effluent (FE). **G** Relative frequencies of contigs either attenuated or persisting partitioned by drug class. Contigs are derived from the resistance gene-MGE gene catalog. The top panel is hospital sewage-associated gene contexts, and the bottom is that associated with background municipal sewage. Points are individual genes, and the distribution reflects the number of contigs associated with each gene in the respective categories. A corresponding plot with individual genes is provided in Supplementary Fig. 12. Over the course of the experiment, the levels (as reflected i.e., peak areas from UPLC/MS/MS suspect screening of PPCPs) of (**H**) clarithromycin decreased, (**I**) erythromycin increased, (**J**) azithromycin remained constant, (**K**) erythromycin-anhydrous increased, and (**L**) sulfamethoxazole increased, as reflected by their peak areas. Sampling dates on the x-axis correspond to before the introduction of hospital sewage (03–23), initially after the introduction of hospital sewage (03–30), and two weeks after the introduction of hospital sewage (04–11). All peak areas are scaled by $10^5$. **H**–**L** Numeric values and brackets indicate Wilcoxon-Rank sum test *p* values and groups compared, respectively. No adjustment for multiple comparisons was deemed necessary. For each respective time point (03–23, 03–30, and 04–11) and condition (0% and 10%), *n* = 3. Source data are provided as a Source Data file. All boxplot summary statistics are: center line: median; upper/lower hinges: 75th and 25th percentiles, respectively; upper and lower whiskers represents the data points extending from the hinge to at most 1.5 times the interquartile range.

drug classes, had correspondingly different fates in AS (Fig. 4G). Specifically, there was microdiversity-level variability in the persistence of *mphA*, *msrE*, *tet(39)*, *tet(G)*, *tet(O)*, and *sul1/sul2* and *mer* family resistance genes, among others (Supplementary Fig. 12).

Interestingly, the heterogeneity in gene fate observed here coincided with changes in levels of several antibiotics (Fig. 4H–L, Supplementary Data 4). Antibiotics that increased over the course of the experiment included erythromycin (Fig. 4H–K) (Kruskal-Wallis: *p* = 0.001; Wilcox: median peak area 0 vs. $7.5 \times 10^4$, *p* = 0.005), erythromycin-anhydrous (Kruskal-Wallis: *p* = 0.009; Wilcox: peak area $2 \times 10^4$ vs. $3.5 \times 10^4$ *p* = 0.013), and sulfamethoxazole (Kruskal-Wallis: ns; Wilcox: $2 \times 10^4$ vs. $0.9 \times 10^4$, *p* = 0.04). Increases in erythromycin were concomitant with a nearly 40-fold reduction in median levels of clarithromycin, another macrolide antibiotic, although this was not statistically significant (Fig. 4H). We note that these antibiotic levels were not elevated as a result of the addition of hospital effluent, as the

observed changes occurred in all reactors. Instead, the increases reflect changes in the local sewage used as the background feed. Macrolide prescription rates in outpatient settings have been shown to have winter/early spring peaks[32] suggesting this may have been a byproduct of community antibiotic usage. Therefore, we next addressed the potential consequences of changing antibiotic levels on (1) shifts in the abundance of bacteria harboring the resistance genes, and (2), the potential for HGT of contaminant-associated resistance genes.

## Hosts of resistance genes display widely varied trajectories

If the fate of specific resistance gene contexts were chiefly driven by the persistence of their respective hosts, it would be expected that the relative abundance of the hosts would follow those of the resistance gene contigs. While examination of bin abundances revealed a pattern of change over the duration of the experiment that mirrored

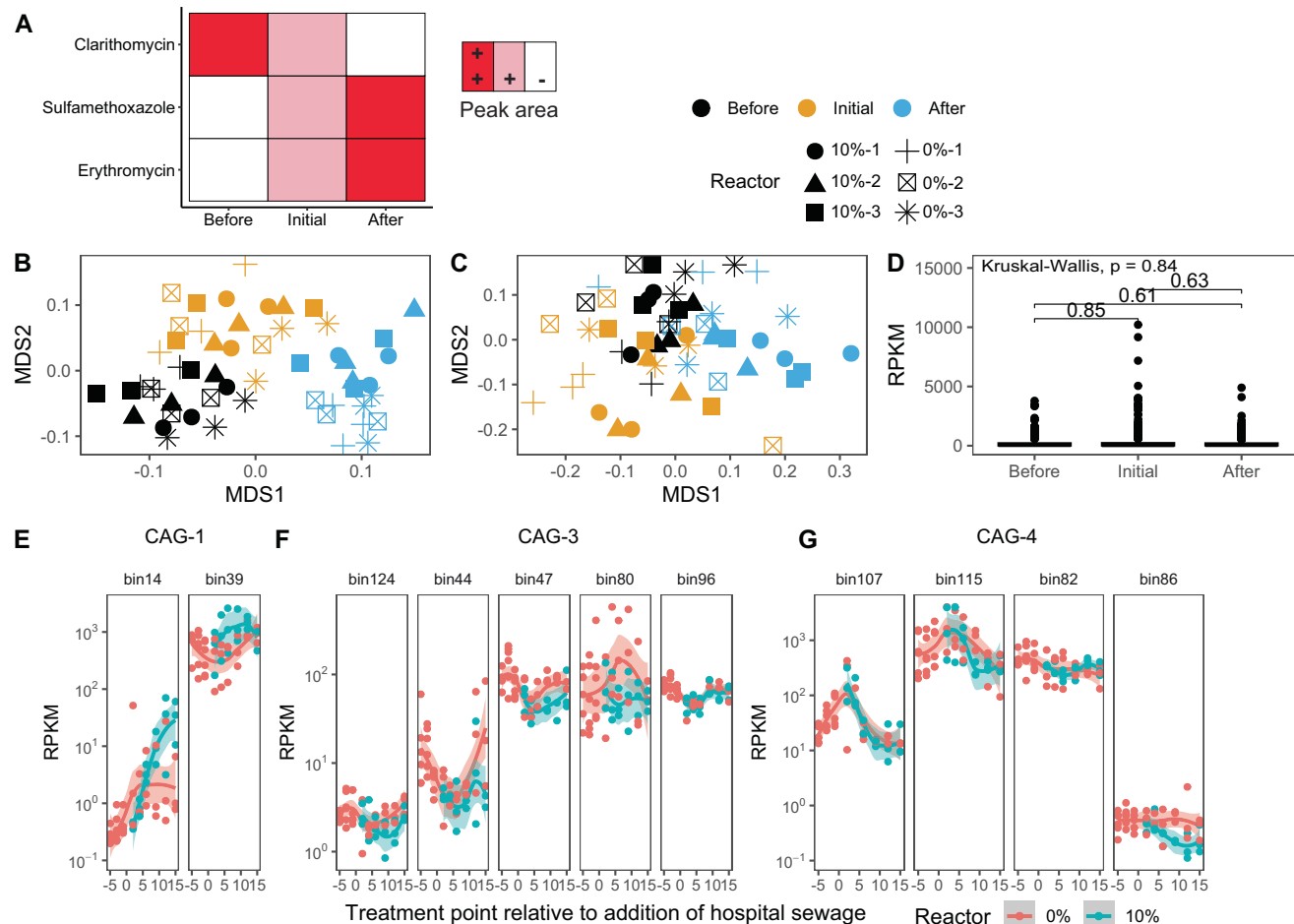

**Fig. 5 | Hosts of macrolide and sulfonamide resistance genes vary in trajectory in the presence of fluctuating levels of the corresponding antibiotics.**
**A** Overview of the patterns in peak areas of identified antibiotics before, initially after, and 2 weeks after introduction of hospital sewage, as detailed in Fig. 4H–K. **B** NMDS of all bin relative abundances across time. **C** Same as (**B**) but subset to only include bins with macrolide or sulfonamide resistance genes. **D** Boxplot of relative abundance of bins bearing macrolide or sulfonamide resistance genes. Numeric values and bars represent Wilcoxon-rank sum p-values and comparisons, respectively. Individual points represent samples taken from biological replicate reactors (n = 6 per sampling point). All boxplot summary statistics are: center line: median;

upper/lower hinges: 75th and 25th percentiles, respectively; upper and lower whiskers represents the data points extending from the hinge to at most 1.5 times the interquartile range. **E−G** Relative abundances of different hosts of macrolide or sulfonamide resistance genes partitioned by coabundance group (CAG). **E** CAG-1 relative abundances over the course of the experiment. **F** CAG-3 relative abundances. **G** CAG-4 relative abundances. Before: up to five days before the addition of hospital sewage; initial: the first few days following addition of hospital sewage; after: the last five days of the experiment following addition of hospital sewage. 5B–G: Source data are provided as a Source Data file.

fluctuating antibiotic contaminant levels (Fig. 5A, B), there was no evidence of bulk selection by erythromycin or sulfamethoxazole for hosts of these resistance genes specifically (Fig. 5C, D). However, different putative hosts displayed disparate trajectories (Fig. 5C–G). For example, *Nannocystis* genomospecies (gs.) (bin39) (phylum *Myxococcota*) displayed changes in relative abundance consistent with an enriching effect (Wilcox: median 500 RPKM vs. 1000 RPKM, *p* < 0.001) (Fig. 5E) and *Thauera* gs. (bin96) (phylum *Proteobacteria*) displayed a moderate reduction (Wilcox: median 75 RPKM vs. median 45 RPKM, *p* < 0.001) followed by an increase in relative abundance (Wilcox: median 45 RPKM vs. 75 RPKM, *p* < 0.001) (Fig. 5E–G). By contrast, *Chitinophagaceae* gs. (bin115), *Acinetobacter sp003987695* (bin107), and *Escherichia coli* (bin86) decreased or remained unchanged over the duration of the experiment (Fig. 5G). Whereas multiple bins bearing macrolide resistance genes were found, only one bin, *Dokdonella* gs. 82 (phylum *Proteobacteria*) bearing a sulfonamide resistance gene (*sul2*) was found. The sole MAG with *sul2* displayed decreased relative abundance over time (Wilcox: median 450 RPKM vs. 200 RPKM, *p* < 0.001). Further, relative abundance profiles of persisting or attenuated contigs (as predicted by Kairos) followed expectations in that

attenuated contigs corresponded to hosts that decreased in AS relative to influent and persisting contigs corresponded to hosts with abundances that remained the same or increased (Supplementary Figs. 12, 13).

**Postulated pathways of resistance gene in situ HGT**
It was observed that resistance genes were found across diverse genera, classes, and phyla (Fig. 3A–C), suggesting the potential for HGT. Because of the changes in profiles of antibiotics, we assessed the potential for in situ HGT possibly linked to the antibiotics using the general framework proposed previously[19], with formal and case-specific hypotheses crafted for this experimental design (Fig. 6A, B). In this case, in situ HGT strictly refers to any occurrence of cross-taxa gene sharing with a pattern of presence/absence in samples consistent with an HGT event, or an enrichment of a pre-existing genome bearing the gene, in the sampled period of time.

For this analysis, we applied the Kairos derep-detect workflow to identify contigs for which identical resistance gene or mobileOGs were found, but where different taxonomic assignments were predicted (i.e., gene sharing or potential instances of HGTs). Kairos imposes

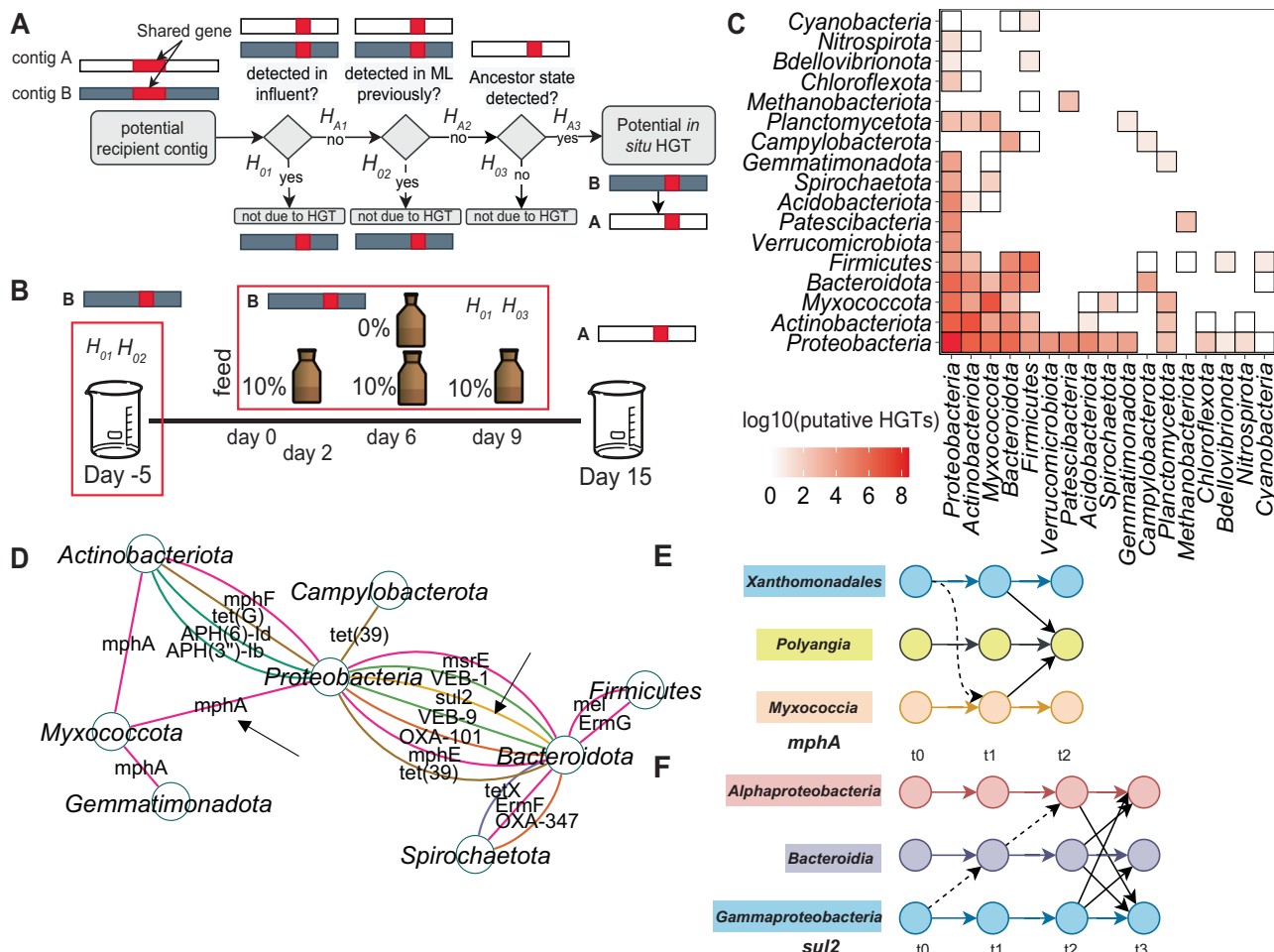

**Fig. 6 | Multiple potential pathways explain putative in situ horizontal transfer of sulfonamide resistance gene *sul2* and macrolide resistance gene *mphA*.**
**A** Criteria used to define in situ HGT using the resistance gene contig catalog and deeply sequenced activated sludge and influent sewage samples. **B** Deeply sequenced samples used in inferring in situ HGT. Red boxes enclose samples used to test the associated hypotheses. **C** Total prevalence of predicted HGT events across phyla. Horizontal is the number of HGT events within a given phyla. **D** Gene sharing graph spanning phyla representing potential HGT events. Arrows indicate connections meeting in situ criteria. **E** Transfer diagram for *mphA* representing potential pathways between *Xanthomonadales* and members of the phylum *Myxococcota* (*Myxococcia* and *Polyangia*). **F** Transfer diagram for *sul2* representing potential pathways between *Bacteroidota*, and *Alphaproteobacteria* and *Gammaproteobacteria*. E&F: dashed lines represent potential, but undetected pathways. 6 C: Source data are provided as a Source Data file.

strict similarity criteria for identifying putative HGT events (minimum 99% amino acid identity and 60% coverage). These were selected to optimize detection of very recent HGT events, particularly those associated with resistance genes. This suggested the potential for extensive HGT across multiple taxonomic levels including phylum ($n = 4919$), (Fig. 6C, Supplementary Fig. 14, Supplementary Data 5), class ($n = 1884$) (Supplementary Data 6), order ($n = 3488$) (Supplementary Data 7), family ($n = 983$) (Supplementary Data 8), and genus ($n = 1400$) (Supplementary Data 9). Genes shared between phyla included resistance genes ($n = 143$) APH(6)-Id, APH(3″)-Ib, OXA-205, *qacH, ermG, ermB, mel, mphE, msrE, mphA, mphF, sul2, tet(C)*, and *tet(39)* (Fig. 6C, D). The majority ($n = 3413$) corresponded to mobileOGs of diverse categories (Fig. 6C). Gene sharing network analysis revealed dense linkages connecting *Proteobacteria* to *Bacteroidota* and *Actinobacteriota*, but not *Actinobacteriota* to *Bacteroidota* (Fig. 6D), These connections were negatively correlated with GC content dissimilarity (Spearman's rank $rho = -0.24$, $p < 0.001$). Of the 12,685 potential HGTs across all taxonomic levels, 1608 met in situ criteria, including transfers of *mphA, sul1*, and *sul2* (Fig. 6D) and 14 other resistance genes (OXA-2, *qacH, sul1*, OXA-205, *merF, merP, merT, qacF, merA, merD, merE, merR2*, and *merT*). We focused subsequent analysis on putative transfers of *mphA* and *sul1/sul2* as these genes were

disproportionately persistent (Fig. 4G, Supplementary Fig. 12) and were most likely to inform the potential impact of erythromycin and sulfamethoxazole.

Analysis of predicted recipients and donors suggested a discrete set of potential transfer pathways (Fig. 6E, F). For *mphA*, a single recipient in the class *Polyangia* (phylum *Myxococcota*) was predicted for two potential donors of the orders *Xanthomonadales* (phylum *Proteobacteria*) and *Myxococcia* (*Myxococcota*) (Fig. 6E, Supplementary Fig. 15). By contrast, the pathway for *sul2* included multiple potential recipients and donors and could variably be described by HGT through some route spanning *Gammaproteobacteria, Alphaproteobacteria* and *Bacteroidia*. The use of the in situ criteria did rule out transfer from *Alphaproteobacteria* to *Bacteroidia*, but not from *Gammaproteobacteria* to *Bacteroidia* (Fig. 6F, Supplementary Fig. 16). A single pathway of *sul1* transfer was detected that suggested a transfer from *Sphingomonadaceae* of *Alphaproteobacteria* to *Gammaproteobacteria*, however the putative donor and recipients aligned against one another at their respective edges (Supplementary Fig. 17).

### Multi-level transfer of *mphA* linked to a novel myxophage
Further scrutiny of the *Myxococcota* genetic contexts of *mph*A revealed that they likely were derived from a phage (Fig. 7). This

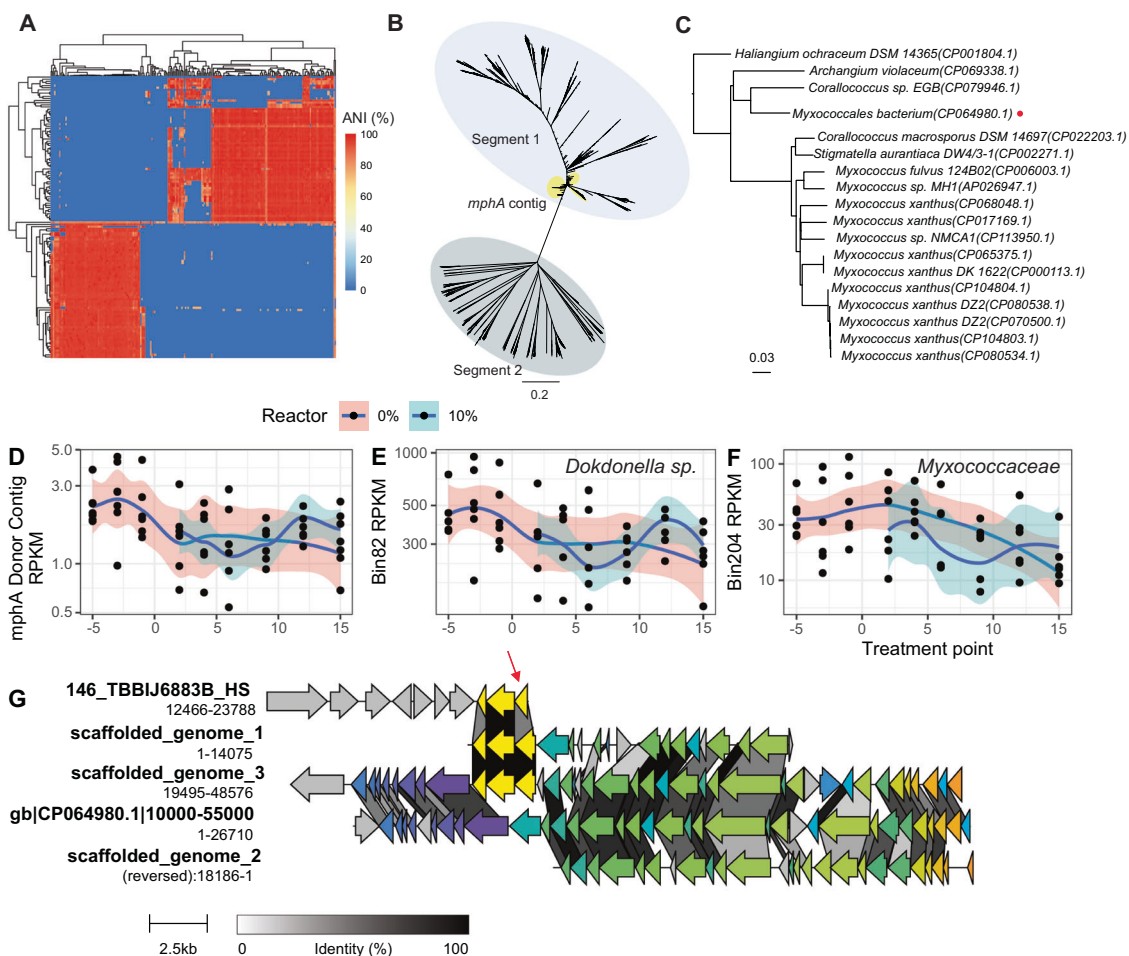

**Fig. 7 | HGT of *mphA* encoded by a novel myxophage highlights complex ecological interactions shaping resistance gene fate. A** Average nucleotide identity heatmap of phage-associated contigs highlights two distinct segments. **B** Mash distance phylogeny of the phage contigs. Yellow circles are those bearing copies of *mphA*. **C** Distribution of similar putative prophage regions across multiple members of the *Myxococcota* phylum. Red circle indicates selected genome for comparison in (**G**). **D** RPKM abundances of putative *mphA* donor contig in

*Proteobacteria*. **E** Abundance of *Dokdonella* sp. MAG bin82 over time is similar to that of the putative donor contig. **F** Abundance of one of the *Myxococcia* MAGs, bin204 has similar trends to that of the *Dokdonella* sp. **G** Genome organization around *mphA* in the donor bacterial sequence and the recipient phage contigs/ scaffolded genomes. Yellow genes are *mphA* and its proximal genes, a *tetR* type transcriptional regulator and a gene encoding an MFS efflux pump. Red arrow indicates *mphA*. 7**D**–**F**: Source data are provided as a Source Data file.

finding is notable, as the role of phages in the evolution of antibiotic resistance remains unclear, especially in environmental matrices. Further inspection revealed that closely-related phage genomes were detected in many samples, but the assemblies were fragmented into two approximately 20,000 base pair segments (Fig. 7A), essentially dividing the genome into halves, with only a few of the fragments encoding *mphA* (Fig. 7B). Draft genomes were constructed by scaffolding contigs based on their alignment to a similar prophage region in NCBI (Supplementary Methods S1, Supplementary Fig. 18) (Fig. 7D). The assemblies produced by HybridSPAdes and OPERA-MS predicting the encoding of *mphA* in the phage genome were validated by the identification of nanopore reads that aligned to both *Myxococcota* genomes and *Enterobacterales* in NCBI. The alignments corresponded to the prophage and *mphA* coding region, respectively (Supplementary Data 10, Supplementary Figs. 19, 20). Similar putative prophage regions were detected in numerous publicly available genomes (Fig. 7C, Supplementary Table 1), but lacked any known resistance genes. However, one putative prophage detected in Danish AS-derived MAG CP064980.1 (Fig. 7D) was integrated near genes encoding macrolide ATP-binding transporter/permease, MacA/MacB (Supplementary Fig. 19). Relative abundances of putative donors and recipients, and the taxonomic assignment of contigs, suggested that the *Proteobacteria* context was possibly associated with bin82, a *Dokdonella* sp.

The contig had remarkably similar abundance profiles and concordant taxonomic assignments (Fig. 7F, G). The potential donor of the predicted class-level HGT of *mphA*, taxonomic assignment *Myxococcia*, was represented by multiple contigs (Supplementary Fig. 15). Only two MAGs of class *Myxococcia* were found, which displayed opposite trends in abundance over time (Fig. 7H, I). Potential recipients of *mphA* encoded by the myxophage (according to class-level taxonomic assignment ascribed to the phage contig) included 43 *Polyangia* dereplicated bins (Supplementary Data 11) spanning all four CAGs.

While the precise pathway of HGT is uncertain, the fact that one of the *Myxococcia* MAGs displayed trends in abundance similar to those of the *Dokdonella* sp. suggests that there may have been an ecological linkage between the two populations represented by the bins (Fig. 7E, F). Additionally, one of the putative recipients displayed a genetic context suggestive of prophage integration. Members of the phylum *Myxococcota* span a wide range of different environments, including sewage, soil, and marine environments[33,34]. Most, but not all, have been demonstrated to have some degree of predation[35]. We examined genomic evidence of a predatory lifestyle (e.g., secretion systems and antibiotic biosynthesis pathways[36]) in the genome of the putative donor, *Archangium* gs. (bin204), of the class *Myxococcia* (Fig. 7G). Functional annotations suggested the presence of partial or complete type 1 secretion systems (T1SS), T2SS, T3SS, T4SS, and T7SS;

at least one antibiotic biosynthesis monooxygenase; one polyketide synthase; and 19 separate CAZy classified carbohydrate active enzymes (Supplementary Data 12).

## Discussion

The hospital sewage that was the subject of this study was found not to substantially impact the treatment performance, taxonomic composition, or resistome of the AS reactors, a topic that is explored in greater depth in a parallel study[30]. Here, we integrated analysis of short- and long-read metagenomic sequencing, PPCP screening, and HGT analysis using Kairos, our recently developed bioinformatics software package in order to investigate the impact of fluctuating antibiotic levels on the fate of resistance genes. Mobile resistance genes derived from both background municipal sewage and hospital sewage were largely attenuated by AS treatment, but a subset of resistance genes persisted and were correlated, in part, with a shift in the levels of macrolide and sulfonamide antibiotics (Fig. 4). We identified dense networks of gene sharing within and between phyla, including the putative transfer of *mphA* within *Myxococcota* via a novel myxophage, and at least two compelling instances of hypothetical in situ HGT. This assessment was achieved through use of a lab-scale AS system seeded with field-collected AS and municipal and hospital wastewater feeds. Integrated analysis of short- and long-read metagenomic sequencing, PPCP screening, and HGT analysis using Kairos, our recently developed bioinformatics software package, made these observations possible.

The findings here illustrate a striking example of where residual antibiotics appear to act as selective agents for the proliferation of AR genes during wastewater treatment. However, the potential impact of antibiotic contamination was not apparent using low-resolution metagenomic analyzes, such as short read alignment to reference databases or taxonomic classification[37]. Alternatively, by integrating hybrid assembly, MAGs, and microdiversity-aware sequence analysis, we identified a dynamic shift in the microbiome over the course of the experiment due to HGT (Figs. 6, 7) and both increases and decreases of specific bacterial lineages predicted to be hosts of the relevant resistance genes (Figs. 5E-G, 7F-K).

It is acknowledged that the in situ criteria used here cannot differentiate between clonal enrichment of a rare genome bearing the putatively transferred gene vs. an HGT event. The in situ criteria could be met either through HGT that occurred at some previous time (and was in sufficiently low abundance to elude detection) and then amplified by host-level selection, or via recombination that occurred during the sampled period. We also note that it is unlikely antibiotics directly stimulated transfer, as, thus far, there are few examples of direct biochemical activation of MGE-associated recombination[38]. By contrast, selective enrichment of specific variants, some of which bear horizontally-acquired genes, is more parsimonious, and has previously been shown to be a key factor in the emergence of resistant phenotypes[39]. While it is mechanistically important to distinguish between direct induction of recombination or selection for a stochastic event, both result in elevated copies of the putatively transferred gene. In this case, it was particularly notable that the emergence of the putative HGT also co-occurred with elevated antibiotic levels.

There was no evidence of bulk selection for hosts of macrolide- or sulfonamide-resistance genes, again highlighting that the impact of residual antibiotics may be uneven or multifaceted. Profiles of abundances across hosts of macrolide and sulfonamide resistance genes were variable, implying that changes in host abundance were at least not solely due to shifting antibiotic contamination (Figs. 5E-G, 7F-K). Rather, our results suggest that the influence of antibiotics on selection is moderated by pre-existing dynamics among members of the community and possibly via HGT.

Surprisingly, we predicted the in situ transfer of *mphA* between *Proteobacteria* and *Myxococcota* via the activity of a novel myxophage,

a puzzling biological inference. While it is unclear how the phage came to encode *mphA*, the co-occurrence of extensive genetic diversity in the myxophage assemblies, the presence of identical copies of *mphA*, and elevated erythromycin seems too improbable to be a coincidence, although this cannot be ruled out. It is noteworthy that *Myxococcota* have an unusual predatory lifestyle[33] and are able to prey on a broad array of organisms, including both fecal-associated bacteria (which are abundant in sewage, e.g., *E. coli*, *Klebsiella*)[40] and soil-associated bacteria (including *Xanthomonas fragariae*)[35]. While it is possible a predator-prey relationship might explain the association between the *Myxococcota* MAGs and the *Dokdonella* sp. MAGs[41], more experimentation would be necessary to derive such mechanistic insight. Members of the *Myxococcota* phylum have recently been reaffirmed as active predators and important players in wastewater microbiomes[42], and the extensive collection of 52 MAGs presented here should aid in further characterization of their niche in wastewater.

The present work brings to the fore several important observations that are emergent from the literature. On the one hand, phages have been suggested to contribute to resistance gene mobility in wastewater[43]; however, whether they play substantive roles in resistance gene mobility remains controversial[23]. Here, we found a highly active and diverse myxophage that was prolific among members of the phylum *Myxococcota*. In addition, our analyses suggested the potential for phylum-HGT in a relatively short time scale, defying, at least, our own expectations. However, this observation is consistent with a recent observation that gene-sharing graphs derived from global sewage samples frequently span phyla[44]. One potential explanation may be inferred from a previous bioinformatic investigation of integrons, which indicated that shared environment, rather than phylogenetic background, was most predictive of integron sharing[45]. Regardless, as shown here, transduction in conjunction with additional modes of mobility (such as transformation) may facilitate the mobility of genes via an ecologically distinct mechanism relative to conjugation.

Wastewater is increasingly being recognized as a potential source of novel resistance genes due to its coalescence of sub-clinical levels of antibiotics, extreme genetic diversity, and contact with natural environments[10,46]. While the emergence of new resistance genes is likely to be extremely rare[10], factors governing this process remain uncertain. As presently postulated[47], the emergence of novel ARGs can be generally described as being driven by three factors: means, motive, and opportunity. A potential novel resistance determinant must be mobilized out of its original non-resistance context and subject to HGT (means); then, the gene must be enriched or persist in recipient organisms, likely due to selection by antibiotics or cross-selection with other selective agents (motive). Finally, physical proximity to organisms undergoing selection must coincide with the mobilization (opportunity). Here, we found evidence that reaffirmed this model, albeit for previously characterized ARGs. More broadly, our findings highlight the potential for interactive effects of selective agents, microbial ecology, and HGT in the evolution of antibiotic resistance in the environment, including the emergence of novel resistance determinants.

## Methods

### Sequencing batch reactor design and operation

Two sets of triplicate SBRs were operated for ~three months in three-liter glass beakers with an active volume of two liters in a temperature controlled room, as described in Maile-Moskowitz et al.[37]. The SBRs were operated on a 12-hour cycle with a two-day hydraulic retention time and five-day solids retention time. Each cycle consisted of a 10.78 h aeration period, including a 60-minute feed, followed by 8 minutes of solids wasting (decanting of AS), 53 minutes of solids settling, and a 12-minute effluent decant. During the aeration/react period the SBRs were aerated using Top Fin® Aquarium Air Pumps and

mixed using stir plates. SBRs were fed and decanted using three- or four-roller peristaltic Masterflex® EasyLoad pump heads controlled by Masterflex® pump drives (Model 7553-80). Influent feed was obtained from a small, local Virginia WWTP (average flow of 3 million gallons per day), while hospital sewage was obtained from an urban medical centre in Chicago, Illinois. Untreated hospital sewage was collected from manholes over a 24-hour period and shipped to the Virginia Tech lab on ice. Upon arrival, hospital sewage was stored at 4 °C for 56 days prior to commencing the experiment. We were unable to obtain fresh hospital sewage prior to the commencement of the dosing as the experiment occurred in February 2020, just before global COVID-19 stay-at-home orders were emplaced. After reaching steady-state, SBRs were maintained for 17 days post-hospital sewage addition.

## Shotgun metagenomic sequencing

Mixed ester cellulose filters were washed with autoclaved nanopure water before wastewater samples (AS, influent or effluent) were filtered through 0.22 μm mixed ester cellulose filters in triplicate until clogging of the filter occurred, with approximate volumes and water weight recorded for each sample. A lab blank was also included for each sampling point which received only autoclaved nanopure water. To capture potential contamination incurred during sample preparation, DNA extraction, and sequencing, we added 37 μL of the Zymo Mock Microbial Community (DS6700, Zymo Research, Irvine, CA) whole cell spike into the lab blank filter tube prior to extraction. For sequencing, a representative sample of about 10 lab blanks was combined and submitted for sequencing in parallel for each flow cell.

DNA was extracted using the MP Bio spin-kit for soil (MP Biomedicals, Irvine, CA) with the following modifications. We increased the first centrifugation step by 10 minutes to increase the separation of filter fragments from soluble supernatant. Final elution was conducted in molecular-grade water. DNA was quantified using a qubit fluorometer with the high-sensitivity dsDNA detection assay kit from Thermo Life Sciences (Q33120, Thermo Fisher, Waltham MA). Samples were then submitted to the Duke University Center for Genomic and Computational Biology for library preparation with the KAPA Hyper-Prep kit and sequencing on an Illumina NovaSeq6000. Nanopore sequencing was performed on pooled samples from conditions (e.g., 10%-1, 10%-2, and 10%-3 day 1) using a nanopore minION sequencer (Oxford Nanopore Technologies). Library preparation was performed using the ligation sequencing kit SQK-LSK109 with native barcoding (NBD-104) following manufacturer's protocol (vNBE_9065_v109_revJ_23May2018) and loaded onto an R10.3 flowcell. Reads were basecalled using guppy v.3.2.10 and a minimum q-score of 7 was imposed.

## Short read preprocessing and sequence analysis

Paired end metagenomic reads were quality filtered and decontaminated using bbduk (ktrim=r k = 23 mink=11 hdist=1 tpe tbo maq 4). Decontamination included removal of adapter sequences, the JGI contaminant database[48], and a custom database of sequences derived from a sample of negative controls. Quality filtered and decontaminated reads were queried against CARD v3.0.7[49], and experimental sequences in BacMet v2[50]. Resistance genes were annotated at a minimum identity of 80% and $e < 10^{-10}$ using diamond[51]. Resistance gene counts were normalized to 16 s rRNA copies derived from bowtie2[52] mapping of short reads against GreenGenes[53] v13.5 (-x 1000--very-sensitive) and reads per kilobase million (RPKM). Taxonomy was annotated using kraken2 with gtdb v202 as the underlying taxonomy database.

## Statistics and Reproducibility

Analyzes were performed in R v. 4.1. No statistical method was used to predetermine sample size. No data were excluded from the analysis. The experiments were not randomized. The Investigators were not blinded to allocation during experiments and outcome assessment.

## Assembly, co-assembly, binning and dereplication

Multiple hybrid assembly strategies were performed using short Illumina reads and long minION nanopore reads to improve recovery of informative resistance gene contexts. Briefly, individual samples were assembled using OPERA-MS[54] (--contig-len-thr 1000 −long-read-mapper minimap2) and hybridSPAdes[55] (metaspades.py with default settings). OPERA-MS was used for all coassemblies, including individual reactors (e.g., 10%-1) across all timepoints, coassembly of all ML samples, and of samples partitioned by treatment (i.e., ± hospital effluent or 10% vs. 0%). MAGs were generated from each of the assemblies/coassemblies in the following way: MAGs were predicted from coassemblies by first aligning short reads from corresponding samples to the assembly (using both bbmap and minimap2 in separate runs) and then binning using MetaBat2[56] and MaxBin[57]. Individual sample assemblies were mapped with only the original sample using both MetaBat2 and MaxBin using minimap2[58] -x sr and bbmap. The resulting MAGs were dereplicated using derep[59] v. 2.1 with default settings except with an adjusted minimum contamination cutoff of 5%. CAGs were defined using a correlation matrix derived from RPKM abundances of all bins in AS samples only. Correlations were calculated in R using cor(method = "spearman") and converted to a distance matrix using vegdist(method = "euclidean") from vegan v2.6-4, followed by hclust(method = "complete"). Clusters were picked based on the dendrogram and the Dunn index. Final abundance estimations for the collection of MAGs were performed by aligning short reads to the MAGs using bowtie2 (-x 1000--very-sensitive) and relative abundances were extracted using samtools[60] coverage. Taxonomic assignments for the MAGs were determined using gtdb-tk[61].

## Annotation of mobile resistance genes and MGEs

All assemblies/coassemblies were searched for resistance genes and MGE hallmark genes. Protein sequences were predicted using prodigal[62] (-meta) and queried against experimental sequences in BacMet v2, CARD v3.0.7, and mobileOG-db beatrix-v1.6[63] using diamond blastp (-id 90% -e 1e-10). For subsequent contextual analysis, only those contigs with a hit from one of the databases was retained. MGE marker hits were subclassified into element classes of plasmid (sequences derived from COMPASS[64] or NCBI Plasmid RefSeq[65]), transposable element (sequences derived from ISfinder[66]), integrative (sequences derived from ICEberg[67] and integration/excision category proteins not included in ISfinder), or conjugative types (sequences with the transfer major mobileOG category and conjugation minor category) using the script getElementClassifications.R (https://github.com/clb21565/mobileOG-db/blob/main/scripts/getElementClassifications.R) on the beatrix-v1.6 metadata file.

## Analysis of HGT with Kairos

To construct gene-sharing networks and identify potential HGTs, Kairos[19] derep-detect was used. Kairos derep-detect identifies near identical (≥ 99% identity) proteins in contigs with different taxonomic classifications (determined here using mmseqs2[68] with gtdb[69,70] v202 as the underlying database). To assess support for the presence of variants (in this case as represented by highly similar contigs produced through multiple assembly/co-assembly strategies), Kairos assess input reads to short windows extracted from two contigs corresponding to a variable region. The windows are defined by a length $l$ (where $l = 75$ bp, by default) to the left and right of the bounds of the aligned regions in both contigs (150 bp total). These edges were then dereplicated using mmseqs2 (--min-seq-id 0.99 -c 0.88 --cov-mode 1). The 88% coverage criterion was empirically found to improve performance[19]. Reads are aligned to the extracted windows with an imposed minimum alignment length of 100 bp by default, ensuring

that at least 25 bp of the region of variation that is unique to that insertion is represented in the aligned region. By including both windows in the read alignment step, reads are directly compared to the two similar loci simultaneously, thus reducing the likelihood that reads from one will erroneously map to the other. Contigs are deemed present if 90% or more of the distinguishing loci are detected. To identify MAGs that were associated with the putative HGTs observed, we compared their taxonomic annotations to those of the contigs in the resistance gene-MGE catalog.

## Suspect Screening of PPCPs

All water samples were pre-filtered through 0.7 μm glass fibre filters (Whatman, Maidstone, UK). Triplicate samples (200 mL each) were then extracted for the target analytes, cleaned up of background matrixes, and finally concentrated using solid phase extraction (SPE). The final extracts were then qualitatively screened for the presence of a total of 138 PPCPs on an ultra-performance liquid chromatography-tandem mass spectrometry (UPLC/MS/MS) using the method described in (Supplementary Methods S2). The multi-compound screening approach employing UPLC/MS/MS used a custom-made compound identification database and was semi-quantitative. This approach enables the calculation of relative change based on peak areas of a screened compound in samples among those tested within one analytical batch of samples analyzed. Because quantification for all 138 compounds that were screened for in all water samples was prohibitively expensive and time-consuming, all samples collected over the course of the experiment were extracted, cleaned up, and analyzed within one analytical batch to ensure the tracking of a compound's relative change over time by comparing its peak area in the samples.

## Reporting summary

Further information on research design is available in the Nature Portfolio Reporting Summary linked to this article.

## Data availability

All sequencing reads have been deposited to the SRA under BioProject PRJNA1020581. MAGs can be accessed at https://zenodo.org/records/10028566. Source data are provided in this paper.

## Code availability

Scripts associated with the present work are available here: https://github.com/clb21565/metagenomics/tree/main/HospitalEffluentProject. Current versions of Kairos can be found here: https://github.com/clb21565/kairos.

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

## Acknowledgements

The authors thank Callen Barlik and Monica Gurung for their assistance in processing samples and Just Stein for help with metagenomic binning. The authors acknowledge Advanced Research Computing at Virginia Tech for providing computational resources and technical support that have contributed to the results reported within this paper. URL: https://arc.vt.edu/ Icons used in figures were licensed under Creative Commons License 3 and 4. Specifically, Fig. 1a used genome-sequencer-5 icon by DBCLS https://togotv.dbcls.jp/en/pics.html is licensed under CC-BY 4.0 Unported https://creativecommons.org/licenses/by/4.0/; Fig. 6b and Supplementary Figs. 5 and 10 used brown-bottles-2d icon by OpenClipart https://openclipart.org/ is licensed under CC0 https://creativecommons.org/publicdomain/zero/1.0/; and Figs. 1a, 6b, and Supplementary Figs. 5 and 10 used beaker-empty icon by Servier https://smart.servier.com/ is licensed under CC-BY 3.0 Unported https://creativecommons.org/licenses/by/3.0/. We acknowledge National Science Foundation (NSF) Awards #1545756 (PV), #2004751 (LZ) and #2125798 (AP) and Water Research Foundation Project 4813 (AP). The research presented was not performed or funded by EPA and was not subject to EPA's quality system requirements. The views expressed in this article are those of the author(s) and do not necessarily represent the views or the policies of the U.S. Environmental Protection Agency.

## Author contributions

C.L.B. conducted experiments, performed analysis, and wrote and edited the manuscript. A.M.M. conducted experiments and edited the manuscript. A.J.L. contributed to the interpretation and edited the manuscript. K.X. conducted screening test for PPCPs, wrote, and edited the manuscript. L.K.L. helped to conceive the study, performed experiments and contributed to the writing and editing of the manuscript. B.D. helped write and edit the manuscript. L.Z. acquired funding, contributed to the analysis, and edited the manuscript. A.P. and P.V. acquired funding, conceived the experiment, and edited the manuscript.

## Competing interests

The authors declare no competing interests.
