## [Peer Review File · Nature Communications]

Selection and horizontal gene transfer underlie microdiversity-level heterogeneity in resistance gene fate during wastewater treatmentREVIEWER COMMENTS

Reviewer #1 (Remarks to the Author):

The manuscript "Selection, ecology, and horizontal gene transfer under microdiversity-level heterogeneity in fate of resistance genes during wastewater treatment" by Brown et al. is based on the operation of municipal wastewater fed sequencing batch activated sludge reactors with and without inflow augmentation of hospital sewage. They then used a multi-faceted approach consisting of bioinformatic-, culture- and chemistry-based methods to determine the effect of hospital sewage augmentation on SBR performance as well as the spread of AMR in the SBR system, with a specific focus on the complex ecological and evolutionary mechanisms underlying this spread. Within this context, they utilize their recently developed, state-of-the-art KAIROS package to analyse microdiversity differences and assess microbiome-level horizontal gene transfer events of ARGs and their specific mode of transfer.

The general idea behind this study is sound and is worthy of investigation and the manuscript is a clear demonstration of the potential of the new KAIROS package. However, I feel that only half of the results obtained (those identifying novel and potential horizontal gene transfer pathways and modes of transfer) are of high enough interest for the high standards of Nature communications. While that part is of high quality and novelty, the part looking into the augmentation of the SBR with hospital sewage does not produce clear results and suffers from several weaknesses pertaining to the SBR operation and subsequent sequencing protocol. The results of that section are at times inconsistent and the correlation between antibiotics detected and resistance gene fate seems to only hold in specific cases, rather than generally and I see several weaknesses that need addressing specified below. Further the presentation of the results in that part of the manuscript is at times confusing, consequentially masking the really interesting HGT results appearing later which could be on their own of enough interest for a manuscript. Separating the two parts could improve the significance of the manuscript.

In the augmentation experiment the augmentation with 10% hospital sewage is only given based on volumetric ratios. However, the ratio of bacteria between the two waste water types and resistance genes between the two wastewater types might be very different and should be taken into account. Further, the abundance of bacteria and resistance genes in the activated sludge reactor should be taken into account. I suggest to calculate based on the feeding rates and the mixing to perform a microbial mass balance and determine if the observed results are significantly different than what would be expected based on random mixing of the three components as the potentially high bacterial abundance in activated sludge and the high detection limit of metagenomics might have contributed to the proposed loss of e.g. resistance genes and hospital indicator bacteria which might simply be an artifact of mixing and detection limits.

The authors state that their reactors were allowed to reach steady state (e.g., L. 121) ahead of the experimental period. How do they define this? Based on microbiome, resistome, function? Which proportion of the results is hence attributed to the actual feeding with hospital wastewater and which proportion is based on natural evolution of the activated sludge community?

Would the Procrustes test that seems to suggest linkage between diversity and resistome (l.157) hold when removing the undiluted hospital samples as an outgroup, as no treatment-based grouping is visible in the genus level, while treatment-based grouping is visible in the resistome analysis?

I struggle with the authors claim of running their experiment under controlled conditions (e.g L60). Is it really controlled conditions if e.g. the antibiotic concentrations are highly variable in the influent?

The authors state that based on their genus level assessment of diversity "<0.1% abundance in more than half the samples" was considered potential false positives (l.141). However, in the paragraph following this, all genera with statistically significant different abundances between the treatments fall below or far below this threshold. Are these significant differences reliable or potential false positives as well?

Overall figure legends are lacking in details making the figures difficult to assess. For example, in Figure 1 "INF, ML, FE" are not defined, temporal dynamics in Figure 1D are not displayed. In addition the legends often rather describe results than clearly stating which data is shown. Also figure panels referred to in the text regularly don't match the content of the sentences (e.g. L 291 & L294). Please check throughout the manuscript that the right panels are referred to.

References should be presented ahead rather than after the period of the sentence they belong to throughout the manuscript.

How can one see results for single resistance genes in the figure that is referred to (L320)?

Reviewer #2 (Remarks to the Author):

The study by Brown et al., reports on the proliferation of Antibiotic Resistance Genes (ARGs) within a small-scale, laboratory wastewater-treating bioreactor that is supplemented with hospital wastewater. This is an important topic for research because we need to understand better where, and under what conditions, ARGs proliferate in order to prevent the spreading of these genes towards minimizing public health risk. The study provides new insights into this topic such as that some ARGs, very likely, do proliferate within these settings, and this seems to be driven by relatively high concentrations of the corresponding antibiotic as well as stochastic processes. Hence, the study is worthwhile and, I believe, broadly interesting. However, there are alternative explanations for some of the key results reported that the authors need to consider towards better qualifying their statements (or provide more data in support of their own preferred explanation). Most notable is the possibility of rare genomes not detected in the original inoculum vs. horizontal gene transfer (HGT), and mis-assembly of the corresponding genomes vs. phage-mediated HGT. Please see also below for more details. Finally, the authors seem to imply that the mechanisms for ARG proliferation within wastewater facilities remain elusive (e.g., Lines 422-423). However, I am not sure this is really true. I mean, it is a highly anticipated fact that the corresponding antibiotic should be abundant to select for, and thus cause the proliferation of the ARGs (the opposite would have been news!); and indeed, this is what the authors found, by large, I think. So, I think the authors should qualify their statements better, here and elsewhere. That said, the study is worthwhile because it does document such events during wastewater treatment (but the study does not elucidate a novel, or not previously suspected or documented, underlying mechanism. It is also likely that I am confused by the word "elusive" and what the authors really mean by it).

Abstract: Why the ARGs would increase in abundance? Is there selection for them e.g., the antibiotic is present in the hospital wastewater? I think that was indeed the case from some of the results reported later on in the manuscript and I believe it would be helpful, and would make the abstract more complete, to report this information in the abstract.

How are you sure it is HGT and not a rare genome/strain not detected in the original inoculum (or wastewater) due to low sequence coverage of the community becoming abundant? I don't think the authors have proven it is HGT; they have accumulated some evidence in support of HGT but can not prove it unless they sequence every genome from the beginning of the reactor and the hospital inoculum, which is not technically feasible really. So, I would suggest qualifying their statements better by recognizing this alternative scenario more, and up front. Related to this: What was the level of coverage of the microbial community achieved by sequencing? You could use Nonpareil to assess this as an example. In general, giving an idea of how deep the coverage was would be useful, here and elsewhere.

Line 68-71. I don't understand this sentence. Please check.

Lines 294-296 and elsewhere. Might be better to refer to the bins with the term "genomospecies" and their lowest assigned taxonomic or the species name when they have a close match at the

species level (e.g. >95% ANI). It is hard to remember "bin 1", "bin 2" etc (but we can remember a species name or more).

Line 357. Do you mean an integrated prophage or just a lytic virus that was assembled as part of the (host) genome or something else? Please clarify. Could it be that the prophage (?) genome was misbinned as part of a MAG? The authors seem to suggest that they have long-read data that span the genome, and thus support phage-mediate HGT, if I got this right, but there are issues with the latter data and it does not sound that they are conclusive. Specifically, in lines 365-366: is it a diagnostic part of the Myxococcota genome or just maps to another mobile element that is not specific to the genome? Also, the match was to plasmid not prophage, it seems, right? So, how you can conclude it is a prophage? More details are needed here, I think.

Reviewer #3 (Remarks to the Author):

I clarify that my review covers only the liquid chromatography-mass spectrometry analysis. I know that this does not represent the core of the work, however, some changes/clarification are needed.

In Supplementary Methods:

Section S2. Untargeted PPCP analysis

Extraction and cleanup of water samples using solid phase extraction (SPE)

1) Please, add the following:

The complete name of the SPE stationary phase, namely OASIS HLB

Add cartridge dimension (6 mL I guess, or 3 mL?) and stationary phase amount (500 mg? 400 mg? 200 mg?)

2) "screened for 138 PPCPs": therefore, please, here and in the main manuscript, change "Untargeted analysis" with "suspect screening". From the instrumental point of view, differences between the two approaches are negligible, whereas the processing workflow is different for data handling and compound identification. In the untargeted approach, there are no information about the possible detected compounds (as typically encountered in dissolved organic matter or some environmental sample analysis); in the suspect screening approach, the detection of expected/hypothesized compounds is carried out by an inclusion list. Moreover, the instrumental method presented here, using a low-resolution mass spectrometer, is more similar to a targeted analysis approach, since only predefined compounds were searched for.

UPLC/MS/MS screening for PPCPs

4) "coupled with a tandem mass spectrometry (6490...": instrument typology is missing. Was the instrument a triple quadrupole? I guess so, but the authors have to specify.

5) Are the authors sure about column size/flow-rate?

I mean, the column is a classical analytical one for HPLC (not for UPLC, for internal diameter and particle size), namely 4.6 mm × 50 mm, 5 μm. It is also quite short (5 cm only) and it was operated at a medium-high temperature (40°C). Moreover, acetonitrile gives a low back-pressure compared to methanol, for example. Therefore, based on my experience, a 0.3 mL min⁻¹ flow-rate seems too low. Please, check and/or clarify. I was wondering if the in-column flow-rate was higher, and the authors reported the flow-rate after post-column splitting.

6) "of four analytical standards: Carbamazepine, Erythromycin, Sulfamethazine, Chlorotetracycline, Tylosin, Sulfamethoxazole, and Triclosan": please, correct the number of analytical standards.

Main manuscript

7) Antibiotic levels are only reported in Figs. 4-5, if I did not miss them. In my opinion, it would be useful to add in the Supplementary section the list of identified compounds and their peak area values in the different samples.

REVIEWER COMMENTS

Reviewer #1 (Remarks to the Author):

The manuscript “Selection, ecology, and horizontal gene transfer underly microdiversity-level heterogeneity in fate of resistance genes during wastewater treatment” by Brown et al. is based on the operation of municipal wastewater fed sequencing batch activated sludge reactors with and without inflow augmentation of hospital sewage. They then used a multi-faceted approach consisting of bioinformatic-, culture- and chemistry-based methods to determine the effect of hospital sewage augmentation on SBR performance as well as the spread of AMR in the SBR system, with a specific focus on the complex ecological and evolutionary mechanisms underlying this spread. Within this context, they utilize their recently developed, state-of-the-art KAIROS package to analyse microdiversity differences and assess microbiome-level horizontal gene transfer events of ARGs and their specific mode of transfer.

The general idea behind this study is sound and is worthy of investigation and the manuscript is a clear demonstration of the potential of the new KAIROS package. However, I feel that only half of the results obtained (those identifying novel and potential horizontal gene transfer pathways and modes of transfer) are of high enough interest for the high standards of Nature communications. While that part is of high quality and novelty, the part looking into the augmentation of the SBR with hospital sewage does not produce clear results and suffers from several weaknesses pertaining to the SBR operation and subsequent sequencing protocol. The results of that section are at times inconsistent and the correlation between antibiotics detected and resistance gene fate seems to only hold in specific cases, rather than generally and I see several weaknesses that need addressing specified below. Further the presentation of the results in that part of the manuscript is at times confusing, consequentially masking the really interesting HGT results appearing later which could be on their own of enough interest for a manuscript. Separating the two parts could improve the significance of the manuscript.

In the augmentation experiment the augmentation with 10% hospital sewage is only given based on volumetric ratios. However, the ratio of bacteria between the two waste water types and resistance genes between the two wastewater types might be very different and should be taken into account. Further, the abundance of bacteria and resistance genes in the activated sludge reactor should be taken into account. I suggest to calculate based on the feeding rates and the mixing to perform a microbial mass balance and determine if the observed results are significantly different than what would be expected based on random mixing of the three components as the potentially high bacterial abundance in activated sludge and the high detection limit of metagenomics might have contributed to the proposed loss of e.g. resistance genes and hospital indicator bacteria which might simply be an artifact of mixing and detection limits.

Dear reviewer, we are sincerely grateful for your careful reading and evaluation. We agree that the presentation of the hospital sewage results was lacking in full clarity. It was an oversight on our part that we did not inform the reader that many of these important details pointed out by the reviewer are the subject of a companion manuscript that is still in

preparation but is available as a dissertation chapter (Maile-Moskowitz, A., 2023, pp. 47-83). This manuscript includes a closer look at the hospital sewage and its impact on the SBRs. We have also included it in the resubmission as material for review only.

In addition to responding to the reviewer's specific comments below, we have taken steps to reduce the emphasis on hospital vs. municipal sewage comparisons and more thoroughly discuss the companion manuscript. This included but was not limited to removing the differential abundance analysis (L123) and editing the discussion (L385):

Here, ~~(we applied finer resolution tools (...))~~ we integrated analysis of short- and long-read metagenomic sequencing, PPCP screening, and HGT analysis using Kairos, our recently developed bioinformatics software package in order to investigate the impact of fluctuating antibiotic levels on the fate of resistance genes.

However, we judge that these changes do not conflict with our overall finding that hospital effluent did not have a substantial impact on the SBR microbiome and resistome. This point was raised in the original draft as in the following section:

L135-137 of the original draft:

Hospital sewage was found to have only a minor impact on the organic carbon and nitrogen removed by the SBRs and the composition of the corresponding microbiomes and resistomes (Fig. 1A-D).

Because we have reframed the objective of the study to focus more prominently on microdiversity, HGT, and the impact of the fluctuating antibiotic levels, we decided that the mass balance analysis was no longer essential to the present work. This analysis will be included in the companion paper.

The authors state that their reactors were allowed to reach steady state (e.g., L. 121) ahead of the experimental period. How do they define this? Based on microbiome, resistome, function? Which proportion of the results is hence attributed to the actual feeding with hospital wastewater and which proportion is based on natural evolution of the activated sludge community?

We thank the reviewer for pointing out this accidental omission. The operational definition for steady-state applied in this study is now included, e.g. L107:

Until they reached steady state [(i.e., defined in this study as stable removal of organic carbon)],

Would the Procrustes test that seems to suggest linkage between diversity and resistome (l.157) hold when removing the undiluted hospital samples as an outgroup, as no treatment-based grouping is visible in the genus level, while treatment-based grouping is visible in the resistome analysis?

We thank the reviewer for this suggestion and repeated the Procrustes analysis on ordinations made without the undiluted hospital samples and found similar results, albeit with slightly poorer fit (L140):

The resistome was strongly linked to genus-level taxonomic profiles (Procrustes: $m^2 = 0.90$, $p = 0.001$; [excluding undiluted hospital effluent, 0.8 , $p=0.001$]), (Fig. 1D,E), suggesting a strong partitioning of ARGs into separate genera.

I struggle with the authors claim of running their experiment under controlled conditions (e.g L60). Is it really controlled conditions if e.g. the antibiotic concentrations are highly variable in the influent?

Our intention was to convey that the conditions were controlled and replicated relative to a field-scale wastewater treatment plant study. We agree with the reviewer that technically this is not a fully “controlled” experiment and have modified the text accordingly:

L78: Here, we employed sequencing batch reactors (SBRs) for [semi-]controlled simulation

L394: This assessment was achieved through use of a ~~controlled~~ [lab-scale] AS system seeded with field-collected AS and municipal and hospital wastewater feeds

The authors state that based on their genus level assessment of diversity “<0.1% abundance in more than half the samples” was considered potential false positives (l.141). However, in the paragraph following this, all genera with statistically significant different abundances between the treatments fall below or far below this threshold. Are these significant differences reliable or potential false positives as well?

We thank the reviewer for their careful reading. As described above, we eliminated the differential abundance analysis and instead limited the analysis of the hospital sewage in the revised draft and cite the companion manuscript to refer the reader to these details.

Overall figure legends are lacking in details making the figures difficult to assess. For example, in Figure 1 “INF, ML, FE” are not defined, temporal dynamics in Figure 1D are not displayed. In addition the legends often rather describe results than clearly stating which data is shown. Also figure panels referred to in the text regularly don’t match the content of the sentences (e.g. L 291 & L294). Please check throughout the manuscript that the right panels are referred to.

We thank the reviewer for catching the figure legend error and have corrected it. In addition, we proofread the manuscript again to identify and correct cases where the figures being referenced were incorrect.

Specifically:

- Added abbreviations to figure legend of Figure 1
- Corrected the in text references to specific figure panels, particularly in the “Postulated pathways of resistance gene *in situ* HGT section (L288)

And to be more specific about the data shown rather than describe results:

- Added text to figure 2 legend to better explain data presented

- Re-wrote legend of figure 4 G to better explain what data are presented rather than describe results

References should be presented ahead rather than after the period of the sentence they belong to throughout the manuscript.

We thank the reviewer for pointing out this formatting issue. Upon reformatting to meet journal guidelines, this issue was resolved.

How can one see results for single resistance genes in the figure that is referred to (L320)?

This text referred to the incorrect figure. Genes described in the adjacent sentence are available in Supplementary Table S5. We have corrected the error.

Reviewer #2 (Remarks to the Author):

We have re-organized the following reviewer comments according to the specific points raised to provide a clearer response.

The study by Brown et al., reports on the proliferation of Antibiotic Resistance Genes (ARGs) within a small-scale, laboratory wastewater-treating bioreactor that is supplemented with hospital wastewater. This is an important topic for research because we need to understand better where, and under what conditions, ARGs proliferate in order to prevent the spreading of these genes towards minimizing public health risk. The study provides new insights into this topic such as that some ARGs, very likely, do proliferate within these settings, and this seems to be driven by relatively high concentrations of the corresponding antibiotic as well as stochastic processes. Hence, the study is worthwhile and, I believe, broadly interesting.

However, there are alternative explanations for some of the key results reported that the authors need to consider towards better qualifying their statements (or provide more data in support of their own preferred explanation). Most notable is the possibility of rare genomes not detected in the original inoculum vs. horizontal gene transfer (HGT), and mis-assembly of the corresponding genomes vs. phage-mediated HGT. Please see also below for more details.

Finally, the authors seem to imply that the mechanisms for ARG proliferation within wastewater facilities remain elusive (e.g., Lines 422-423). However, I am not sure this is really true. I mean, it is a highly anticipated fact that the corresponding antibiotic should be abundant to select for, and thus cause the proliferation of the ARGs (the opposite would have been news!); and indeed, this is what the authors found, by large, I think. So, I think the authors should qualify their statements better, here and elsewhere. That said, the study is worthwhile because it does document such events during wastewater treatment (but the study does not elucidate a novel, or not previously suspected or documented, underlying mechanism. It is also likely that I am confused by the word “elusive” and what the authors really mean by it).

We appreciate Reviewer 2’s perspective on this important point and are grateful for the words of support. Firstly, we have modified the text to be more specific and to remove

the descriptor “elusive”, which is ambiguous. Secondly, we agree that it is the prevailing and intuitive hypothesis that antibiotics would be expected to promote antibiotic resistance in wastewater. However, to date, there has been surprisingly little evidence to support this hypothesis. In many published cases there is no observable effect of antibiotics in sewage or even negative correlations. For example, as quoted below in a recently published prominent review article (Larsson et al., 2022):

*Although some studies report increases in the relative abundance of certain ARGs in such environments, it is difficult to distinguish whether this is a result merely of taxonomic changes, unrelated to antibiotic selection pressures, or from direct selection of resistant strains within species^{66,67,68}. Although plausible, definite evidence for such direct selection in sewage treatment plants is still lacking, and some evidence points to the opposite⁶⁹. A recent study on sterile-filtered wastewaters indicated no selective effect of the investigated treated municipal effluent and a small selective effect by untreated influent. By contrast, untreated hospital wastewater strongly selected for multiresistant *Escherichia coli* in different controlled exposure experiments with individual isolates and communities⁷⁰. The exact selective agents responsible therein could not be identified, but the relatively high levels of antibiotics in hospital wastewater make them plausible drivers of resistance selection.*

Review article references:

66. Yang, Y., Li, B., Zou, S., Fang, H. H. P. & Zhang, T. Fate of antibiotic resistance genes in sewage treatment plant revealed by metagenomic approach. *Water Res.* 62, 97–106 (2014).

67. Bengtsson-Palme, J. et al. Elucidating selection processes for antibiotic resistance in sewage treatment plants using metagenomics. *Sci. Total Environ.* 572, 697–712 (2016).

68. Manaia, C. M. et al. Antibiotic resistance in wastewater treatment plants: tackling the black box. *Environ. Int.* 115, 312–324 (2018).

69. Flach, C. F., Genheden, M., Fick, J. & Joakim Larsson, D. G. A comprehensive screening of *Escherichia coli* isolates from Scandinavia’s largest sewage treatment plant indicates no selection for antibiotic resistance. *Environ. Sci. Technol.* 52, 11419–11428 (2018).

70. Kraupner, N. et al. Evidence for selection of multi-resistant *E. coli* by hospital effluent. *Environ. Int.* 150, 106436 (2021).

Review article:

Larsson, D.G.J., Flach, CF. Antibiotic resistance in the environment. *Nat Rev Microbiol* 20, 257–269 (2022). <https://doi.org/10.1038/s41579-021-00649-x>

Abstract: Why the ARGs would increase in abundance? Is there selection for them e.g., the antibiotic is present in the hospital wastewater? I think that was indeed the case from some of the results reported later on in the manuscript and I believe it would be helpful, and would make the abstract more complete, to report this information in the abstract.

As the editorial guidelines for the abstract confine it to 150 words, we had to cut most of the detail from the original draft. We welcome further clarifying comments on the new abstract below:

Activated sludge is the centerpiece of biological wastewater treatment, as it facilitates removal of sewage-associated pollutants, fecal bacteria, and pathogens from wastewater through semi-controlled microbial ecology. It has been hypothesized that horizontal gene transfer facilitates the spread of antibiotic resistance genes within the wastewater treatment plant, in part because of the presence residual antibiotics in sewage. However, to date, there has been surprisingly little evidence to suggest that sewage-associated antibiotics select for resistance at wastewater treatment plants via horizontal gene transfer or otherwise. We addressed the role of sewage-associated antibiotics in promoting antibiotic resistance using lab-scale sequencing batch reactors fed field-collected wastewater, metagenomic sequencing, and our recently developed bioinformatic tool Kairos. Here, we found confirmatory evidence that fluctuating levels of antibiotics in sewage are associated with horizontal gene transfer of antibiotic resistance genes, microbial ecology, and microdiversity-level differences in resistance gene fate in activated sludge.

How are you sure it is HGT and not a rare genome/strain not detected in the original inoculum (or wastewater) due to low sequence coverage of the community becoming abundant? I don't think the authors have proven it is HGT; they have accumulated some evidence in support of HGT but can not prove it unless they sequence every genome from the beginning of the reactor and the hospital inoculum, which is not technically feasible really. So, I would suggest qualifying their statements better by recognizing this alternative scenario more, and up front.

We thank the reviewer for this insightful comment, and wholeheartedly agree. We addressed this in the original draft in the following sections (L307-L312):

It was observed that resistance genes were found across diverse genera, classes, and phyla (Fig. 2A-C), suggesting the potential for gene sharing and possibly HGT. Because of the changes in profiles of antibiotics, we assessed the potential for in situ HGT possibly linked to the antibiotics using the general framework proposed previously (25), with formal and case-specific hypotheses crafted for this experimental design (Fig 6A,B). In this case, in situ HGT strictly refers to any occurrence of cross-taxa gene sharing with a pattern of presence/absence in samples consistent with an HGT event, or an enrichment of a preexisting HGT, in the sampled period of time.

And L419-431:

It is acknowledged that the in-situ criteria used here cannot differentiate between selective enrichment of a stochastic HGT versus an induced recombination event. The in-situ criteria could be met either through HGT that occurred at some previous time (and was in sufficiently low abundance to elude detection) and then amplified by host-level selection, or through recombination within- or between-hosts. (...) By contrast, selective enrichment of specific variants, some of which bear horizontally-acquired genes, is more parsimonious, and has previously been shown to be a key factor in the emergence of resistant phenotypes (43, 44). While it is mechanistically important to distinguish between direct induction of recombination or selection for a stochastic event, both result in elevated copies of the putatively transferred gene. In this case, it was particularly notable that the emergence of the putative HGT also co-occurred with elevated antibiotic levels.

However, we added two additional points:

L300: Kairos imposes strict similarity criteria for identifying putative HGT events (minimum 99% amino acid identity and 60% coverage). These were selected to optimize detection of very recent HGT events, particularly those associated with RGs.

In addition, we added the following qualification to the paragraph above to emphasize the reviewer's point:

In this case, in situ HGT strictly refers to any occurrence of cross-taxa gene sharing with a pattern of presence/absence in samples consistent with an HGT event, or an enrichment of a preexisting HGT [or "rare" genome], in the sampled period of time.

Related to this: What was the level of coverage of the microbial community achieved by sequencing? You could use Nonpareil to assess this as an example. In general, giving an idea of how deep the coverage was would be useful, here and elsewhere.

This is an important point. We added Nonpareil estimates of coverage to the following sections (L115-120):

products (PPCPs) over a period of about three weeks. AS and influent samples were sequenced to an average depth of 5 Gbp/sample [(nonpareil coverage 0.5±0.1)] and effluent sequenced to an average depth of 3 Gbp/sample [(nonpareil coverage 0.5±0.1)]. A subset (n = 6) of samples were sequenced deeply (average of 36 Gbp, [nonpareil coverage 0.8±0.10]). A subset of DNA extracts from biological replicate reactors were also pooled by sampling date and sequenced across three nanopore minION flowcells to a target depth of 1.2 Gbp/sample and 9.4 Gbp total after basecalling ($N_{50} = 1.3$ kbp) (Table S1).

Line 68-71. I don't understand this sentence. Please check.

We apologize for the confusion. This sentence:

Other efforts using shotgun metagenomics have provided ... high-dimensional analyses of ARG and MGE abundances.(20, 21) This latter approach is particularly problematic for studying microbiome-level HGT since resistome(22) profiles (such as those generated via dimension reduction analyses) are chiefly driven by taxonomy, suggesting that a much greater degree of biological granularity is required to reveal HGT and corresponding drivers in WWTPs.(23)

Was referring to a methodological approach wherein short reads are aligned to databases of antibiotic resistance genes and mobile genetic elements, and correlation between the resulting estimates are taken as suggestions of potential HGT. For example, reference 21:

Zhu N, Long Y, Kan Z, Zhu Y, Jin H. Reduction of mobile genetic elements determines the removal of antibiotic resistance genes during pig manure composting after thermal pretreatment. *Bioresour Technol.* 2023 Nov;387:129672. doi: 10.1016/j.biortech.2023.129672. Epub 2023 Aug 15. PMID: 37586429.

Our argument was that this is not an effective way to study HGT. However, we recognize the phrasing was confusing and have changed it (L55-59):

Correlation of ARG and MGE abundances through short read mapping is particularly problematic, as changes to gene abundance are driven largely by changes in the abundance of the host bacteria. By contrast, HGT occurs between individual cells, suggesting that a much greater degree of biological granularity is required to reveal HGT and corresponding drivers in WWTPs.²³

Lines 294-296 and elsewhere. Might be better to refer to the bins with the term “genomospecies” and their lowest assigned taxonomic or the species name when they have a close match at the species level (e.g. >95% ANI). It is hard to remember “bin 1”, “bin 2” etc (but we can remember a species name or more).

This is a great suggestion and we have implemented it throughout the revised manuscript as in the following (L272):

For example, Nannocystis genomospecies (gs.) (bin39) (phylum Myxococcota) displayed changes in relative abundance consistent with an enriching effect (Wilcox: median 500 RPKM vs. 1,000 RPKM, $p < 0.001$) (Fig. 5D)

Line 357. Do you mean an integrated prophage or just a lytic virus that was assembled as part of the (host) genome or something else? Please clarify. Could it be that the prophage (?) genome was misbinned as part of a MAG? The authors seem to suggest that they have long-read data that span the genome, and thus support phage-mediate HGT, if I got this right, but there are issues with the latter data and it does not sound that they are conclusive. Specifically, in lines 365-366: is it a diagnostic part of the Myxococcota genome or just maps to another mobile element that is not specific to the genome? Also, the match was to plasmid not prophage, it seems, right? So, how you can conclude it is a prophage? More details are needed here, I think.

We apologize for the lack of clarity. Multiple nanopore reads were found that aligned to the phage genome and to *mphA*. These reads were then searched against NCBI using the megablastn web server (pictured in the Fig S21 of the original draft):

*Fig. S21. Example nanopore read alignment to both Myxococcota and Enterobacterales genomes. The nanopore read here (6de60e4c-c959-472c-93f4-1fe535ef4dd0) encodes *mphA* as well as phage genes. Alignment details are provided in Extended Data 1.*

We also provided extended alignment details in *Extended Data 1* including an example fastq read:

```
@6de60e4c-c959-472c-93f4-1fe535ef4dd0 runid=f4c844d6bbe51d691623b437e8130a7833033316 sampleid=2020_09_05_ORANGE_327-410
read=64680 ch=253 start_time=2020-09-07T13:46:15Z barcode=barcode06
CGATGTCCTCGTTCAGTCTATCTGTTGCTAAGGTTGAGACTACTTTCGCTTGGGAGAACAGCACCTCATCGGCGTGACGGACCTTTCGGCAACGTCGTCGTCGGCCCTACGAC
CGCGCGAGCTCCGGCCGGTTCGCCGCCCGCGCCGCGGACCCGCGCTTCGACCTGTGGCGCATGCTGCCGAAGCACAACCGACGAGCGACCCGACCGGACCTGTTC
GGTTCATCGCTGCTCGAGAGGTGACGGACCTTCTGCTCGCGCAGCTGACCGCTGGCCGACGCTTCTTCGATCTGGGGAGCGCGCGCCGAGGCGCTTCATCGATCTCATCCT
CGCGCATCTCGCAACCCATTCCCGTTTCGAGCTCGACCTGCTGGCAAGCGCTCTCTGCGCTCGGTCGTCGAGATGTACCGACAGAAAGGCGCGCAAGGATATCC
AGAACGGCATCCGCTTCTTCTCGGCATCGACATCTCCGCCATCACGCCGTTTCAACTCACGCCAACGTTTTCAGCTCGCCCTCGGGGGTCCGAGCTGGCGCTCGACTGGTGTCTCG
CCCCCTCGACCGCTTTCGCGCGCTACGCTTCAAGCTGTTGCGCGCTATCTCAGCGGACCGGAAGCGTCCGCGAGCTCCGGGCCATCGTCGTCGCTGAAGCCCGG
CACACGCACTTCTGGGACCTCGTCGAGCCGCTGCCGCCATCGTGCCGAACCACTGGGAGCTGGGCTCAGCGACCTGGGAAACACGGACCTGCATTGAGGGGGCCGGCGTCGTCAG
GAGTGGTGTATATAACCGACGCGACTGTATAAATTCGCCGAGATCCGTCGATCTCTGGAGCGATTCATGACCGCAGCGTAACCGTCGATACCTCCCACTGCACGC
GCTCGCGCGGATGGGCTGGTTTCATGGCCACTGACCGTGAATGAGCTTGGCTCGACTATCGGTCGTCGATTCGCCCGCTCGACGATGGACGTCGGTGGTGTGCGCATCCCGCGC
CGCGGAGAGTGAGCGGAGGTCGAGCCAGAGGCGGGTGTGGCGATGCTCAAAGCGCTTGGCTTCGCGGTCGCGGACTGGCGGTCGCGCAACGCCGAGGTCGTTG
CCTATCCCATGCTCGAGGACTCGACTCGCATCTCCCTCAACTGGCTCGCTCGCGCGGCTGGACTTCGCGGAGAGCTTCGCGACGCGGCTCGCGCTCGC
GCCGTTCCGTCCTCGCGCGCTAGATGCGGGGATGCTCATCCGACGCGCGCGGAGGCTCGGAGGTCACGACGAGCTTGGAGCGCTCCGACGCGGATTCGTTGGTGA
CGACAAGCGCTCCACCGATGGCAGCGCTGGCTCGACGACGATTCGTCGTCGCGCGGATTCCTGCTGCTGGTGGCTACGTCGGGATCTTACGTCGGCCATGTGCTCGCAAC
ACGGAGCGGCTCAGCGGATGATCGACTGGAGCGAGCGCGGAGTTGGAGATGACCTCCCATCGACATGGCCCTCGCACCTCATGCTCTTTGGCGAGGCGGGCTCA
CTTCTCTCAGCTACGAAAGCGGCTGGAGGGTGTCTTTCGGCGAAGCGAAGTGTCTTAACCTTAGCGATCGTCGCCGACTG
+
% (/) %', (. '4$388#+$0$$$$* (, (85)?=?@B?31140; ) 7899; ) 2, 9*%$#&-25+85/<=@%A%?CBDADD5=>E?E?;>; *5>, +--/
.., 72.; ?AD?; 74; 88; @<>608; --40; -55&245) 0 ($-9; ; 2277?<5+0+//1-
4-FBBDCHKGGE?>G8JAA@CC>E<<EEB>A69?3; ; B>E=942+)) 1') *7861&, ) *%68; 5F?B
BGDGBCB<D>?A@DCE; ; IC=?>B; E@<9=<<<777B: %*65B<<?8>@>A94588$>: 960%&4-9. ( ) +3<; 55 (+, 4557%-/, 2 (, *. 97A=GAA@) &&/** (55.-
/ : 8?=@@AA>; ??AA=?=?<C; JF?<?HHA<C<<EC4GE: @CDDCGG?@EE8E9CA?==: 8DC@=E>D<C@DB<DADB?G7BCBH; CJ>EDA>IFCBBEF7/44?DDED9E
>FGRD@@@KH@BND<=FGOF<E<?B?>HFC=8; 5@>?>IGCEFAAB<@A@2*1) %& ($+'-%*%-
*# (, 71>8) : @8@5) (&$#%* ., 1/+ : 1EH@++<A@>*>; <7<<GB==; @@6; >5; GBKAGCA=95@A=HADEFH?LIAGEC==EED@?=?AA554/*#30313; ; *8+6--
-/@I=C?F>@<CG=C<: 824&%) %$&; ; <?1B30C=CB?
?@BBCDGGJ=B@78F: ==: EFDF@?>D?DD=: @D9@//4: ACH<; FD<; HG6?; @?>A) B?3; >596//><A:<37#3#2%&=: :<@:$, *8?/*878=9+, ==E>8E8>AD@B
B@: ?>7<10: 0, &44=B; DA; @C?<=BEIA>: 885528: & (& (7226@B: @EGB<>@DHCC?>EHPFGBE?>@A>GJH=FEHHE<GFBFA=978>4>A=CADEJI; @>
G?D=1/72: 88763. 7645) %$23499: 598; 8<ADG?<BEFFDBLHL=>?>; >CAA=<93 (&E>46@=>=@251/4=@<<F?ID?@E@>B>=8=>?>>?>?ECBC=@?CB) 3
(-614$%$%$&23?<<<I@DD<==53903; ; DEB<=DJAB<F?99962$%235. /; <>C>A?BGGEI>H?>AA: A; <FFABB; ; CC>EABG?=>=<; 5) G78
76<524?/+?B?F?>9<=6-)+$<9: <: =BCA@<C: 7E>BBA?A?D; =5=: ; >0: 2>9-
>>?>?<203228 (%31: 998*58, 62=C?DI; DCEG: 9) 554; =<=9/<4* ( ) ##%&6527, @0*/; 7665?7... : 757-
5; 0: .99+44223899>6B?AC: A@C878C; BB2<6=<*% .&#+, (-27: 8459=A3597G3=; >: ?@=?>CGDCE@?<9AAD@>
9<A=CB0) @D==8@GIDGFCCB@BD94>78@: 6/: 7) ; , >>@98: 757 (21) /$) &#, '#, /# '$ $' ' & ( (&%-
, $' % . @F7=: 9: 3; ; 0; 432BG: BDCBDC@C; D?DDCB>>G: A=<97C9: **; 5EBAF@B@E?8?: 9>47?; 500/6%, 4/0&#&' (3$/-+*+--
4550'&79; =?1E?3?A@D@BFB@00, (, <855?BGA<C>; 86B?5333 %
$%42.; 59=AA@?7?'3CD@?E@DHD<<: 9?<46-*.$%$%$385+$ (&=, 56$0@; 7: 1-(1) 4$&%, , %$%$*$##$) "
```

The text has been modified to make this analysis clearer:

(L350) *The assemblies produced by HybridSPAdes and OPERA-MS predicting the encoding of mphA in the phage genome were validated by the identification of nanopore reads that aligned to both Myxococcota genomes and Enterobacterales in NCBI. The regions corresponded to the prophage and mphA coding region, respectively (Supplementary Data 10, Supplementary Fig. 19).*

Methods: To identify MAGs that were associated with the putative HGTs observed, we compared their taxonomic annotations to those of the contigs in the RG-MGE catalogue.

Reviewer #3 (Remarks to the Author):

I clarify that my review covers only the liquid chromatography-mass spectrometry analysis. I know that this does not represent the core of the work, however, some changes/clarification are needed.

In Supplementary Methods:
Section S2. Untargeted PPCP analysis

Extraction and cleanup of water samples using solid phase extraction (SPE)

1) Please, add the following:

The complete name of the SPE stationary phase, namely OASIS HLB
Add cartridge dimension (6 mL I guess, or 3 mL?) and stationary phase amount (500 mg? 400 mg? 200 mg?)

We sincerely thank the author for their comments. Information has been added accordingly in Section S2 of SI document

2) “screened for 138 PPCPs”: therefore, please, here and in the main manuscript, change “Untargeted analysis” with “suspect screening”. From the instrumental point of view, differences between the two approaches are negligible, whereas the processing workflow is different for data handling and compound identification. In the untargeted approach, there are no information about the possible detected compounds (as typically encountered in dissolved organic matter or some environmental sample analysis); in the suspect screening approach, the detection of expected/hypothesized compounds is carried out by an inclusion list. Moreover, the instrumental method presented here, using a low-resolution mass spectrometer, is more similar to a targeted analysis approach, since only predefined compounds were searched for.

Response: Point well taken. Revised accordingly in the manuscript and the SI document.

UPLC/MS/MS screening for PPCPs

4) “coupled with a tandem mass spectrometry (6490…)”: instrument typology is missing. Was the instrument a triple quadrupole? I guess so, but the authors have to specify.

Response: Revised accordingly

5) Are the authors sure about column size/flow-rate?

I mean, the column is a classical analytical one for HPLC (not for UPLC, for internal diameter and particle size), namely 4.6 mm × 50 mm, 5 µm. It is also quite short (5 cm only) and it was operated at a medium-high temperature (40°C). Moreover, acetonitrile gives a low back-pressure compared to methanol, for example. Therefore, based on my experience, a 0.3 mL min⁻¹ flow-rate seems too low. Please, check and/or clarify. I was wondering if the in-column flow-rate was higher, and the authors reported the flow-rate after post-column splitting.

Good catch about the type of analytical column listed and thank you for catching this mistake. The correct information should be: “Agilent Zorbax Eclipse Plus C18 analytical column (2.1x100 mm, 1.8 µm)”. Revised in the SI document accordingly. We also found mistakes in describing mobile phase make up and gradients, all corrected in the SI document. Apologize for the mistakes.

6) “of four analytical standards: Carbamazepine, Erythromycin, Sulfamethazine, Chlorotetracycline, Tylosin, Sulfamethoxazole, and Triclosan”: please, correct the number of analytical standards.

We thank the reviewer for catching this and have corrected it in the supplementary methods section.

Main manuscript

7) Antibiotic levels are only reported in Figs. 4-5, if I did not miss them. In my opinion, it would be useful to add in the Supplementary section the list of identified compounds and their peak area values in the different samples.

We did not report the absolute concentration levels of antibiotics in Figs. 4-5. Instead, UPLC/MS/MS peak areas were reported. Figure captions of Figs 4 and 5 were revised to make this point clearer. The identified compounds and their peak area values are included as Supplementary Data 4.

** See Nature Portfolio's author and referees' website at www.nature.com/authors for information about policies, services and author benefits.

REVIEWERS' COMMENTS

Reviewer #1 (Remarks to the Author):

The authors have responded well to the majority of my comments (Reviewer #1) and the revised manuscript is far improved in clarity and focus. I believe that this revised manuscript fulfills the majority of necessary characteristics for acceptance. I do however have two minor comments remaining:

1. I appreciate that the authors have decided to outsource parts of the hospital ww effect into an accompanying manuscript. However, when referring to the companion study "Consistent with the results of a companion study focused on relating SBR operational conditions to higher-level annotation of ARGs and taxonomy³⁰, ...", the reference is not towards the accompanying chapter submitted by Brown et al., but rather to some PhD thesis of a student not involved in this study or part of the accompanying chapter: "Maile-Moskowitz, A. Shotgun metagenomic analysis of antimicrobial resistance in wastewater. (2023)." Can this either be clarified or corrected or in best case replaced by a citation to a preprint of the accompanying chapter?

2. Figure 2 is barely used in the new version of the manuscript, so I suggest merging with figure 3, however, this is not essential.

Reviewer #1 (Remarks on code availability):

Code seems to be complete and a readme file that allows easy navigation is provided.

Reviewer #2 (Remarks to the Author):

The revised article by Brown et al., represents a much-improved version compared to the first submission. The authors have addressed all my major concerns and I believe most -if not all- of the comments of the other reviewers too. I have only a few minor comments remaining. Specifically:

Line 119. N50 is used for assembly quality typically. Is this really meant to be N50 of assembly or the average read length of the long-read sequencing? Seems too low N50 for long-read data (I noticed that the sequencing effort was not large) unless read length is less than that? Is the latter the case? Please check and clarify.

Line 295. "or an enrichment of a preexisting HGT (or "rare" genome)..." indeed, possibly it was an old HGT but we do not know how long ago (could be quite old), and I think call it HGT may be somewhat confusing. I think it is better to just call it "enrichment of a pre-existing genome that already possessed the gene" (without mentioning HGT) because the key here is to contrast this scenario with the scenario that HGT occurred during the experiment (operation of the reactor), right?

Lines 406-407. "It is acknowledged that the in situ criteria used here cannot differentiate between selective enrichment of a stochastic HGT versus an induced recombination event." I believe this sentence, and likely the next couple sentences, may be confusing because successful HGT events are mediated by recombination (homologous or non-homologous) so (successful) HGT without recombination is not possible, and thus the two parts of the sentence are essentially saying the same thing instead of being contrasting (as "versus" clearly indicates). Like above, I suggest HGT vs. clonal enrichment of a rare genome already carrying the gene before the treatment (and just "clonal enrichment" after that point, or something like this).

Reviewer #3 (Remarks to the Author):

The manuscript has been revised following all the reviewers's comments. For the scientific part within my competence, I do not have any further suggestions.

Reviewer #3 (Remarks on code availability):

The manuscript has been revised following all the reviewers's comments. The scientific part within my competence, namely the liquid chromatography-tandem mass spectrometry section, has been revised and corrected

REVIEWERS' COMMENTS

Reviewer #1 (Remarks to the Author):

The authors have responded well to the majority of my comments (Reviewer #1) and the revised manuscript is far improved in clarity and focus. I believe that this revised manuscript fulfills the majority of necessary characteristics for acceptance. I do however have two minor comments remaining:

1. I appreciate that the authors have decided to outsource parts of the hospital ww effect into an accompanying manuscript. However, when referring to the companion study "Consistent with the results of a companion study focused on relating SBR operational conditions to higher-level annotation of ARGs and taxonomy 30, ...", the reference is not towards the accompanying chapter submitted by Brown et al., but rather to some PhD thesis of a student not involved in this study or part of the accompanying chapter: "Maile-Moskowitz, A. Shotgun metagenomic analysis of antimicrobial resistance in wastewater. (2023)." Can this either be clarified or corrected or in best case replaced by a citation to a preprint of the accompanying chapter?

We apologize for any confusion and have updated this citation to be the direct chapter citation (former draft reference 37). This is the dissertation chapter by co-author Ayella Maile-Moskowitz that reported an analysis of the reactor metagenomes. We have also updated the reference to include the DOI of the dissertation so it can be accessed more readily.

2. Figure 2 is barely used in the new version of the manuscript, so I suggest merging with figure 3, however, this is not essential.

We conferred amongst the co-authors of the manuscript and ultimately decided to keep figure 2 separate from figure 3.

Reviewer #1 (Remarks on code availability):

Code seems to be complete and a readme file that allows easy navigation is provided.

Reviewer #2 (Remarks to the Author):

The revised article by Brown et al., represents a much-improved version compared to the first submission. The authors have addressed all my major concerns and I believe most -if not all- of the comments of the other reviewers too. I have only a few minor comments remaining. Specifically:

Line 119. N50 is used for assembly quality typically. Is this really meant to be N50 of assembly or the average read length of the long-read sequencing? Seems too low N50 for long-read data (I noticed that the sequencing effort was not large) unless read length is less than that? Is the latter the case? Please check and clarify.

We sincerely appreciate the reviewer's input and apologize for the confusion. We have added some additional text to clarify. In short, the nanopore read length was low because we used the same extracts recovered for short read metagenomics (extracted using a bead-beating lysis method). As is noted, this is not a large sequencing effort for the nanopore data, however, it nevertheless turned out to be invaluable for identifying additional contextual information about ARGs associated with the reactor. Moving forward, we have pivoted to specialized extraction kits and Promethion sequencing to improve read length and throughput.

Line 295. "or an enrichment of a preexisting HGT (or "rare" genome)..." indeed, possibly it was an old HGT but we do not know how long ago (could be quite old), and I think call it HGT may be somewhat confusing. I think it is better to just call it "enrichment of a pre-existing genome that already possessed the gene" (without mentioning HGT) because the key here is to contrast this scenario with the scenario that HGT occurred during the experiment (operation of the reactor), right?

This is a good point; we see now that the phrasing is confusing and have updated it accordingly:

In this case, in situ HGT strictly refers to any occurrence of cross-taxa gene sharing with a pattern of presence/absence in samples consistent with an HGT event, or an enrichment of a pre-existing genome bearing the gene, in the sampled period of time.

Lines 406-407. "It is acknowledged that the in situ criteria used here cannot differentiate between selective enrichment of a stochastic HGT versus an induced recombination event." I believe this sentence, and likely the next couple sentences, may be confusing because successful HGT events are mediated by recombination (homologous or non-homologous) so (successful) HGT without recombination is not possible, and thus the two parts of the sentence are essentially saying the same thing instead of being contrasting (as "versus" clearly indicates). Like above, I suggest HGT vs. clonal enrichment of a rare genome already carrying the gene before the treatment (and just "clonal enrichment" after that point, or something like this).

We thank the author for pointing this out. Actually, we meant that it is unlikely that the antibiotics would directly stimulate recombination as is the case for integrative conjugative elements regulated by the SOS-response. In retrospect, we agree that this is confusing. We revised the section as below:

It is acknowledged that the in situ criteria used here cannot differentiate between clonal enrichment of a rare genome bearing the putatively transferred gene vs. an HGT event. The in situ criteria could be met either through HGT that occurred at some previous time (and was in sufficiently low abundance to elude detection) and then amplified by host-level selection, or via recombination that occurred during the sampled period. We also note that it is unlikely antibiotics directly stimulated transfer, as, thus far, there are few examples of direct biochemical activation of MGE-associated recombination.³⁸ By contrast, selective enrichment of specific variants, some of which bear horizontally-acquired genes, is more parsimonious, and has previously been shown to be a key factor in the emergence of resistant phenotypes.³⁹

Reviewer #3 (Remarks to the Author):

The manuscript has been revised following all the reviewers's comments. For the scientific part within my competence, I do not have any further suggestions.

Reviewer #3 (Remarks on code availability):

The manuscript has been revised following all the reviewers's comments. The scientific part within my competence, namely the liquid chromatography-tandem mass spectrometry section, has been revised and corrected

We thank the reviewer for their response and consideration.